# MoonCast: High-Quality Zero-Shot Podcast Generation

**Zeqian Ju**[1,2]     **Dongchao Yang**[3]     **Kai Shen**[2]     **Yichong Leng**[2]     **Zhengtao Wang**[2]

**Songxiang Liu**[2]     **Xinyu Zhou**[2]     **Tao Qin**[4]     **Xiangyang Li**[1]     **Jianwei Yu**[2,†]     **Xu Tan**[2,†]

## Abstract

Recent advances in text-to-speech synthesis have achieved notable success in generating high-quality short utterances for individual speakers. However, these systems still face challenges when extending their capabilities to long, multi-speaker, and spontaneous dialogues, typical of real-world scenarios such as podcasts. These limitations arise from two primary challenges: 1) long speech: podcasts typically span several minutes, exceeding the upper limit of most existing work; 2) spontaneity: podcasts are marked by their spontaneous, oral nature, which sharply contrasts with formal, written contexts; existing works often fall short in capturing this spontaneity. In this paper, we propose MoonCast, a solution for high-quality zero-shot podcast generation, aiming to synthesize spontaneous podcast-style speech from text-only sources (e.g., stories, technical reports, news in TXT, PDF, or Web URL formats) using the voices of unseen speakers. To enable long audio generation, we employ a language model with parameter, data, and context scaling to process sequences in an innovative format designed for modeling entire multi-speaker, multi-turn speech interactions. To enhance spontaneity, we observe that ASR transcripts capture spontaneous speech details (e.g., filler words indicating hesitations, and specific punctuation and spaces reflecting breathing pauses), suggesting that these transcripts can serve as a partial indicator of speech spontaneity. Building upon this assumption, we utilize a script generation module to generate scripts incorporating these spontaneous elements. Experiments show MoonCast outperforms baselines, with notable improvements in contextual coherence and spontaneity.

## 1   Introduction

Recently, significant advancements in large language models (LLMs) and speech codec technologies have substantially enhanced the performance of text-to-speech (TTS) synthesis, improving its naturalness, expressiveness, and tonal richness. These advancements have led to widespread adoption in industries such as customer service and short video production. As TTS technology continues to evolve, there is a growing demand for generating long-duration podcast content from text-only sources, such as news and technical reports. Podcast speech requires not only extended audio lengths but also highly spontaneous expressions, often involving multiple speakers and dynamic interaction.

The limitations of previous efforts in generating high-quality podcasts stem from two key challenges. First, long-context audio modeling presents challenges. Podcasts typically span over several minutes, featuring numerous utterances from multiple speakers. This requires the system to generate not only realistic individual speech but also smooth transitions between utterances. Furthermore, high-quality

[1]University of Science and Technology of China  [2]Moonshot AI  [3]The Chinese University of Hongkong  [4]Microsoft Research.   Correspondence to:  Jianwei Yu <tomasyu@foxmail.com> and Xu Tan <tanxu2012@gmail.com>.

39th Conference on Neural Information Processing Systems (NeurIPS 2025).

podcast generation must account for the contextual coherence of each speaker, encompassing aspects such as prosody and timbre. Second, podcasts are highly spontaneous, typified by the fluid and casual flow of human conversation. They often contain human-like details, including filler words such as "um", occasional hesitations, and minor mistakes. In the multiple-speaker scenario, the system must also account for the interactions between speakers. However, the TTS community has largely focused on improving short individual utterance generation, with limited efforts exploring long-context, spontaneous scenarios. Specifically, academic research has primarily focused on short conversational speech [Nguyen et al., 2023, Mitsui et al., 2023], but these efforts often face difficulties when applied to longer, more complex podcast scenarios, particularly in capturing spontaneity and naturalness within inter-sentence interactions. Recently, industrial solutions like NotebookLM[1] have emerged to facilitate podcast creation from various knowledge sources. However, these solutions often lack transparency in their technical details, limiting their adaptability.

To overcome these limitations, we propose a high-quality podcast generation system MoonCast. On one hand, to improve contextual coherence, we enable the holistic, zero-shot generation of multi-speaker, multi-turn conversations, supported by an innovative sequence format. To manage such extensive context, we adopt a language model-based speech modeling approach, scaled with approximately 500k hours of training data, 2.5B parameters, and a 40K token context length. In addition, we employ a chunk-wise autoregressive speech detokenizer for effective inference in the long-context scenario.

On other hand, to improve spontaneity, we build upon a novel observation: certain spontaneous speech details are often reflected within their corresponding automatic speech recognition (ASR) transcripts, such as hesitations linked to filler words, breathing pauses marked by specific punctuation or spaces, and non-verbal sounds identified as onomatopoeic words. This observation prompts our novel assumption: **ASR transcripts can act as a partial proxy for speech spontaneity.** We further validate this assumption through an experiment designed to identify the impact of varying script spontaneity on a fixed text-to-speech model. The results show that the presence of spontaneous details in the script significantly impacts the spontaneity of the generated speech. This validated assumption informs our core design principle: **train the model to generate spontaneous speech conditioned on ASR transcripts, and during inference, use input text designed to emulate ASR transcript characteristics to elicit spontaneous output**. Accordingly, our audio modeling module is trained on a large-scale dataset of spontaneous speech from diverse sources, with corresponding ASR transcripts generated through a data preparation pipeline. A three-stage curriculum learning approach is also employed to facilitate robust training on spontaneous speech data and progressively equip the model with zero-shot, long-context, and spontaneous speech generation capabilities. Finally, in our script generation module, we provide the LLM with demonstrations and detailed instructions to help it emulate ASR transcript style and incorporate spontaneous elements into the scripts.

With this design, we can generate a spontaneous podcast of up to ten minutes from text-only input sources in a comprehensive manner. The experimental results show that the proposed system consistently outperforms the concatenate baselines in terms of intelligibility, coherence, and spontaneity for multi-lingual podcast generation. Specifically, MoonCast achieves subjective evaluation improvements of $0.39$ in spontaneity, $0.28$ in coherence, $0.05$ in intelligibility, $0.13$ in speech quality and $0.25$ in speaker similarity for Chinese, and $0.68$ in spontaneity, $0.62$ in coherence, $0.15$ in intelligibility and $0.05$ in speech quality for English podcast generation. We invite readers to listen to audio samples at `https://mooncastdemo.github.io` for a more intuitive experience.

**We open-source MoonCast, including the prompts[2] for script generation and the audio modeling module[3] for speech generation, to support future research.**

## 2 Background

### 2.1 Zero-Shot TTS

Zero-shot text-to-Speech synthesis aims to synthesize speech that mimics the characteristics of a target speaker using only a brief prompt speech, without requiring additional fine-tuning [Shen et al., 2023,

---

[1]`https://notebooklm.google.com/`
[2]Refer to Appendix E
[3]`https://github.com/jzq2000/MoonCast`

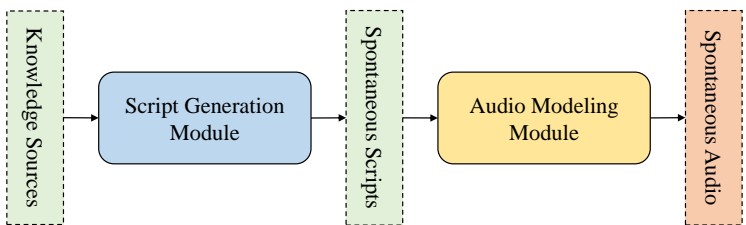

Figure 1: The overall pipeline of the proposed system.

Ju et al., 2024, Chen et al., 2024]. Recent advancements in zero-shot TTS can be broadly categorized into two types based on the representation: discrete code or continuous latent. For the code-based method, VALL-E [Wang et al., 2023] utilizes neural codec language models and achieves high fidelity in zero-shot TTS. Seed-TTS [Anastassiou et al., 2024] and CosyVoice [Du et al., 2024a,b] leverage a single semantic codebook to reduce the difficulty on discrete code generation. Also, discrete diffusion can be leveraged to enable code generation in a non-autoregressive manner [Borsos et al., 2023, Ju et al., 2024]. For the latent-based method, Naturalspeech 2 [Shen et al., 2023] leverages latent diffusion to predict the latent of a speech codec conditioned on a short prompt speech. VoiceBox [Le et al., 2024] utilizes flow matching to model Mel-spectrogram of a speech. In this paper, we adopt a pipeline that combines both the code-based and latent-based methods. Rather than focusing on speech that contains only a single speaker, we consider zero-shot two-speaker podcast speech generation.

## 2.2  Dialogue and Conversation Generation

Most works in zero-shot TTS focus on the speech synthesis of one speaker. However, many scenarios such as dialogues, conversations, and podcasts require the TTS model to be able to synthesize speech with multi-speaker at the same time. Generating spoken dialogues [Schuller et al., 2013] that include natural turn-taking, laughter, and other paralinguistic cues [Zhang et al., 2020, Adiwardana et al., 2020, Lewis et al., 2020, Xu, 2021] is non-trivial. Recent work has explored various approaches to address this challenge. DGSLM [Nguyen et al., 2023] uses a dual-tower transformer architecture to capture the turn-taking dynamics and non-verbal vocalizations in spoken dialogues, aiming to generating naturalistic spoken dialogues. Built on the top of dGSLM, CHATS [Mitsui et al., 2023] makes the generated dialogues more interactive and fluid by incorporating backchannels, laughter, and smooth turn-taking. To enable dialogue generation with diverse timbre, CoVoMix [Zhang et al., 2024] proposes zero-shot dialogue generation to support zero-shot, multi-speaker, multi-round dialogue speech generation. Thanks to the progress in large language model [Achiam et al., 2023, Yang et al., 2024], we can generate speech dialogue with more spontaneous content [Lu et al., 2025]. Despite these advancements, most prior works rely on datasets of approximately 2000 hours (such as the Fisher dataset [Cieri, 2004]) and are limited to generating dialogues of less than 90 seconds. These limitations stem from the challenges in maintaining coherence and naturalness over longer contexts. In this paper, we address these limitations by proposing a long-context text-to-semantic autoregressive architecture to model the inter-sentence prosody, speaker change, and paragraph-level spontaneity.

## 2.3  Spontaneous TTS

Spontaneous TTS refers to the synthesis of speech that mimics natural, conversational speaking styles, as opposed to more formal or read speech. It aims to generate speech with characteristics such as filler words (e.g., "um" and "uh"), diverse rhythms, and natural prosody variations [Yan et al., 2021, Li et al., 2024b]. SponTTS [Li et al., 2024a] proposes a neural bottleneck to help TTS model better model and transfer spontaneous style. Other works [Li et al., 2024b] utilizes LLM to systematically categorize the spontaneous behaviors and then uniformly model these behaviors in TTS model. BaseTTS [Łajszczak et al., 2024] finds out that the spontaneity can come from emergence. Once the TTS model has been trained on a large number of speech data [Anastassiou et al., 2024], it can acquire emergent abilities, such as expressing emotions. Along this direction, in our paper, we further find out that the spontaneity of the generated audio is significantly influenced not only by text-to-speech modeling but also by the script text itself.

# 3 Method

## 3.1 Overall

In this section, we describe the method for podcast speech generation. Conceptually, a podcast consists of multi-speaker, multi-turn spoken dialogues in a spontaneous manner. Unlike traditional zero-shot speech synthesis methods that focus on a single speaker and fixed textual inputs, we divide the podcast generation process into two stages: 1) spontaneous script generation, which converts input knowledge sources into spontaneous text for podcast creation, and 2) spontaneous podcast speech generation, which involves multiple speakers and turns following the generated script. In this paper, we focus on generating two-speaker podcasts.

The overall system pipeline is illustrated in Figure 1. To generate a spontaneous podcast, we first employ an LLM-powered podcast script generation module to produce podcast scripts from input knowledge sources. Subsequently, we utilize a long context audio modeling module to generate podcast speech according to the scripts, using unseen speakers' voices. In specific, for the novel task of podcast speech generation, we represent each audio frame as a single discrete semantic audio code, thereby decomposing the task into text-to-semantic generation and semantic-to-audio reconstruction sub-tasks, using the discrete semantic code sequence as the intermediate representation. As shown in Figure 2, we use a speech semantic codec for speech tokenization, a text-to-semantic model for semantic code modeling, a flow-matching based speech detokenizer for semantic-to-mel reconstruction, and a pre-trained vocoder for mel-to-waveform reconstruction.

# 4 Method

## 4.1 Audio Modeling Module

### 4.1.1 Long-Context Two-Speaker Text-to-Semantic Model

The zero-shot two-speaker podcast generation task aims to synthesize each turn of the podcast using the corresponding speaker's voice, based on the provided reference speech from two speakers. A significant challenge arises from the length of speech code sequences. For example, with a 50 Hz single-layer speech codec (i.e., using a single code to represent a 20 ms speech frame), a common codec setting, a 5-minute podcast corresponds to a sequence length of 15,000. Additionally, unlike single-speaker zero-shot speech synthesis, this task must also ensure contextual coherence and smooth transitions between individual speech segments.

We holistically model the multi-speaker, multi-turn podcast to ensure superior contextual coherence. To effectively manage the extensive context, we utilize a language model-based speech modeling approach. This model is scaled with approximately 500,000 hours of training data, 2.5 billion parameters, and a 40,000-token context length. By inputting raw ASR transcripts as conditioning for speech modeling, we facilitate the model's ability to learn spontaneous patterns directly from their natural occurrence within the conversational speech transcripts used for training.

**Sequence Design.** We design an innovative sequence format for the novel task of zero-shot podcast generation, handling multi-speaker, multi-turn interactions while preserving the continuity of speech as a coherent whole. To achieve this, we adopt a full-text-to-full-audio interleaving approach, rather than interleaving on a per-turn basis. Specifically, as shown in Figure 2, we merge adjacent segments from the same speaker to ensure alternating turns between speakers, and incorporate a special speaker change token after prompts and each podcast turn. This token indicates the change of speaker, thereby enhancing speaker robustness. Formally, we denote the prompt speech codes by $\hat{s}^i$ and the corresponding text by $\hat{t}^i$ for speaker $S^i$, where $i \in \{1, 2\}$. The podcast is represented as a list $[(spk_j, t_j, s_j)]$ consisting of $M$ dialogue turns, where $spk_j$, $t_j$ and $s_j$ correspond to the speaker, the script text and the speech for the turn $j$, where $j \in \{1, \ldots, M\}$. To construct the two-speaker data sequence, we start from creating four sub-sequences by prepending the speaker identifier to each prompt and podcast turn: prompt text $\mathcal{T}^P = \{S^1, \hat{t}^1, S^2, \hat{t}^2\}$, prompt speech $\mathcal{S}^P = \{\hat{s}^1, \hat{s}^2\}$, podcast text $\mathcal{T} = \{spk_1, t_1, \ldots, spk_M, t_M\}$, and podcast speech $\mathcal{S} = \{s_1, \ldots, s_M\}$. These sequences are concatenated in the order $\{\mathcal{T}^P, \mathcal{T}, \mathcal{S}^P, \mathcal{S}\}$. We use the language model to estimate the probability $p(\mathcal{S}|\mathcal{T}^P, \mathcal{T}, \mathcal{S}^P)$. During training, we compute the average cross-entropy loss for each turn, inclduing

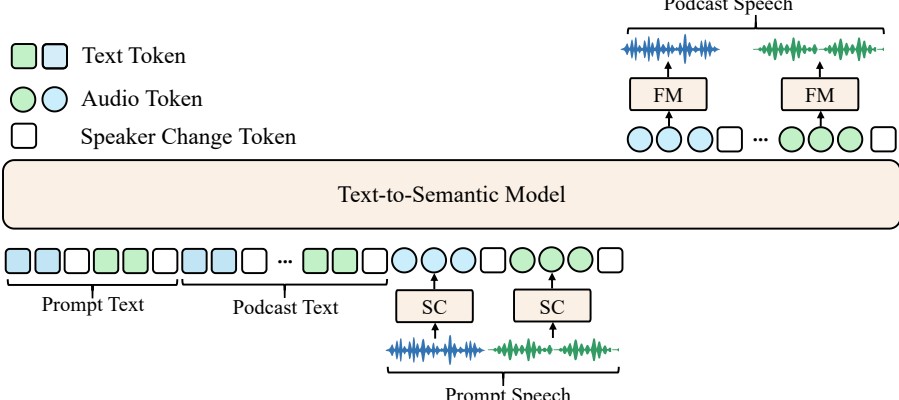

Figure 2: Overview of the audio modeling module. Speaker 1 is represented by blue, and speaker 2 by green. 'SC' denotes the speech Semantic Codec, and 'FM' denotes the Flow-Matching based speech detokenizer.

both speech codes and the special speaker change token. During inference, the predicted speaker-change token allows us to select the appropriate prompt speech for reconstructing the speech.

**Curriculum Learning.** Given the limited availability of high-quality long-form spontaneous speech data, we employ the curriculum learning technique to progressively enhance the model's capabilities across three distinct stages, gradually increasing the complexity of the training data to optimize learning efficiency. 1) In the first stage, we segment the entire audio from all data sources according to the annotations. Each segment contains a single-turn utterance from only one speaker. Specifically, we do not explicitly specify speech prompts, but instead implicitly assume that any prefix of the sequence serves as the prompt for the remainder. We train the model on these individual segments to initially develop its zero-shot TTS capability. 2) In the second stage, we begin modeling entire audio sequences involving two speakers and multiple turns. Given that non-conversational scenarios, such as audiobooks, typically involve less interaction between speakers and feature simpler text with fewer spontaneous details, we start from these data sources, which are easier to learn from. Specifically, we use the first turn from each speaker as the prompt and scale the context length to 40,000 tokens (equivalent to 800 seconds in our setting) to accommodate long-context scenarios. This approach aims to enhance the model's consistency in speaker representation and robustness in long-context, two-speaker scenarios. 3) In the final stage, we refine the model's ability to generate spontaneous speech using conversational data from sources such as podcasts, which feature two speakers and multiple turns. These sources are characterized by dynamic, natural interactions between speakers and the presence of spontaneous speech elements, essential for capturing the nuances of real-world conversations. Similar to the second stage, we use the first turn from each speaker as the prompt while maintaining the context length at 40,000 tokens to support long-context modeling. By exposing the model to this type of data, we aim to improve its ability to generate spontaneous speech.

### 4.1.2 Chunk-wise Autoregressive Speech Detokenizer

To decode the generated speech codes and produce the final podcast, several naive approaches may come to mind, each with its own limitations. One approach might involve reconstructing the entire sequence at once. However, this method faces two major challenges. First, waiting for all tokens to be generated can be prohibitively slow, especially for long speech segments typical in podcast scenarios. Additionally, the large memory footprint required for processing such long sequences often exceeds the available GPU memory. Another naive method is to split the speech into fixed-length segments, reconstruct each segment individually, and then concatenate them. While this approach mitigates the memory issue by reducing the sequence length, it introduces a new problem: discontinuities at the boundaries between chunks. These discontinuities can lead to less fluent and consistent speech, as each segment is generated independently without considering the context of adjacent segments.

To address these limitations, we propose a more efficient solution: a chunk-wise autoregressive detokenizer. This method divides the speech tokens into small chunks (e.g., 3 seconds per chunk),

enabling more efficient processing of long speech segments. By processing the sequence in smaller, manageable chunks, we significantly reduce computational overhead and memory requirements. Additionally, we apply a chunk-wise causal mask, which allows each chunk to access the history of previously generated speech chunks. This approach not only improves the fluency and consistency of the generated speech but also ensures more stable boundaries between chunks, effectively addressing the continuity issues that arise from direct chunking.

**Flow Matching Model.** Our detokenizer is based on a DiT [Peebles and Xie, 2023] based flow-matching model, which conditions on speech codes and generates the mel-spectrogram from random Gaussian noise. Firstly, we take the chunk $i$ and all previous chunks $< i$ where the chunk $i$ is for generation and chunks $< i$ is the prompt for clarification ($i \in [0, N]$, where N is the chunk amount). The chunk $i$'s mel-spectrogram is $\mathbf{M}_i$ and speech codes $\mathbf{C}_i$ ($\mathbf{M}_{<i}$ and $\mathbf{C}_{<i}$ for previous chunks $< i$ as well). The flow-matching approach involves the forward process to add noise to the data, and the backward process to remove the noise in reverse. In training, we apply forward process to obtain the noised data $M_i(t) = t * M_i + (1 - (1 - \sigma_{min})t)\hat{M}$ by mixing the sampled gaussian noise $\hat{M} \sim N(0, 1)$ with clean data $M_i$ at timestamp $t \in [0, 1]$, where $\sigma_{min}$ is a hyper-parameter. The flow-matching model $f_\theta$, parameterize by $\theta$, is adopted to learn the mapping $f_\theta(M_i(t)) = dM_i(t)/dt = M_i - (1 - \sigma_{min})\hat{M}$ with the condition $C_i$. In addition, the $M_{<i}$ and $C_{<i}$ are adopted as the prompts for in-context learning. At inference, we start backward process from another sampled Gaussian noise $M_i(0) \sim N(0, 1)$ and recover the clean data through the ODE: $dM_i(t) = f_\theta(M_i(t))dt$.

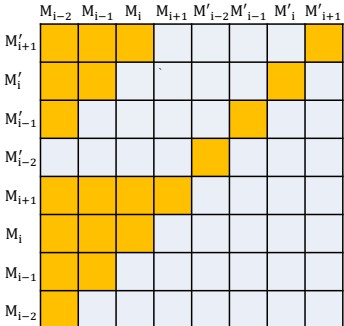

Figure 3: The attention mask design for chunk-wise autoregressive speech detokenizer. $\mathbf{M}_i$ means the clean mel-spectrogram and $\mathbf{M}'_i$ means noisy mel-spectrogram. Yellow means allow to attend, and gray means not allowed to attend. Attention is conducted among row-wise in figure.

**Chunk-wise Causal Mask.** To facilitate efficient training, we first segment the long speech into chunks and apply a chunk-wise causal attention mask [Liu et al., 2024]. This mask allows to access information from both the current noisy and previous clean chunks, thereby enabling the batch-parallel training of all noisy chunks within a single dataloader batch. As shown in Fig. 3, we assume the prompt chunk as $\mathbf{M}_i$ (clean mel-spectrogram) and the chunk for generation as $\mathbf{M}'_i$ (noisy mel-spectrogram). During training, we put all chunks $\mathbf{M}_i$ and $\mathbf{M}'_i$ in a whole sequence, thus there are $2N$ chunks. The attention mask follows: 1) there is no mask in the current chunk; 2) for the left half chunks where $\mathbf{M}_i \in [0, N)$ (i.e., lower rows in Fig. 3), we apply attention mask where $\mathbf{M}_i$ can only attend to the chunks $\mathbf{M}_j$ where $j \in [0, i]$; 3) for the right half chunks where $\mathbf{M}'_i \in [N, 2N)$ (i.e., upper rows in Fig. 3), we apply attention mask where chunk $\mathbf{M}'_i$ can only attend to clean chunks $\mathbf{M}_j$ where $j \in [0, i-1]$ and noisy chunk $\mathbf{M}'_i$ itself.

Following this design, during inference, when the language model outputs a chunk, we use the flow-matching model to generate the corresponding mel-spectrogram $\mathbf{M}$. We then apply prefilling for this chunk with $\mathbf{M}$ and a timestep of $0.999$ to generate the kv-cache for efficient inference.

## 4.2   LLM-Powered Script Generation Module

In this section, we present an LLM-powered podcast script generation module, enabling users to create rich and diverse scripts from different knowledge sources. This module consists of three components: **(1) Content analysis:** For any type of user input (e.g., Web URL, PDF), we combine LLMs to recognize the content in the input. For example, if the user's input is a Web URL, we use the search function in ChatGPT to retrieve the content from the link. **(2) Briefing document generation:**

In our preliminary experiments, we find that directly asking LLM to generate scripts based on the original content often results in ill-suited and vague scripts, which leads to the loss of significant information. To address this issue, we propose generating a briefing document first, which covers the key points in the original content. Specifically, the briefing document includes five components: the title with authors, an abstract, main topics, key citations and a conclusion. Each component includes an additional paragraph to explain technical terms, concepts, or methods that might confuse readers unfamiliar with the field. **(3) Scripts generation:** Based on the briefing document, we use LLM to generate a podcast script that features coherent logic and comprehensive content. Specifically, We guide the LLM in three key areas: podcast structure, format, and content. For structure, we ask the LLM to create engaging openings and closings that set the tone and effectively wrap up the podcast. Regarding format, the script must be in JSON format and feature two speakers: a host who controls the pace of the conversation and a guest who primarily introduces the content of the document. In terms of content, the script includes key citations and explanations of technical terms in a coherent manner, ensuring logical connections between topics and maintaining a moderate information density. To infuse the text with spontaneity and replicate the ASR transcript characteristics in training, we guide the LLM to incorporate spontaneous details such as filler words (e.g., 'um', 'uh', 'like', 'you know', 'so'), response words (e.g., 'yeah', 'right', 'okay'), repetitions and informal grammar. Moreover, we provide formatting tips, such as using spaces and commas within sentences to indicate pauses, and also offer a specific example of an ASR transcript as the demonstration.

# 5 Experiments and Results

## 5.1 Experimental Settings

In this section, we present a overview of the experimental setup, including detailed descriptions of the data preparation, the model architecture, and the evaluation setting.

### 5.1.1 Data Preparation

We conduct our experiments on a large-scale internal Chinese and English audio dataset comprising approximately 1.0 million hours of audio from diverse sources, including podcasts, audiobooks, and audio clips from shows. Following previous works [Yu et al., 2024, He et al., 2024], we apply a data processing pipeline to process these audio source. The final dataset comprises 300,000 hours from Chinese audiobook sources, 15,000 hours from Chinese conversational sources, and 200,000 hours from English conversational sources. Refer to Appendix B for more details.

### 5.1.2 Model Details

**Speech Semantic Codec.** For the speech semantic codec, both the encoder and decoder consist of 12 ConvNext blocks, each with a kernel size of 7 and a hidden size of 384. The 1024-dimensional SSL feature is projected into an 8-dimensional space for quantization using an 8192-entry codebook. We train the codec for 200,000 steps.

**Text-to-Semantic Model** For the text-to-semantic model, we use a 2.5B-parameter, 16-layer Llama-style Transformer with a hidden size of 3072 and 24 attention heads. We train it using the Megatron framework on 64 A100 80GB GPUs with a tensor parallelism degree of 8, over a maximum sequence length of 40k, a batch size of 600, and for 2,000 steps in each curriculum learning stage. We use a top-k value of 30, a top-p value of 0.8, and a temperature of 0.8 for inference. We use Byte-Pair Encoding (BPE) for text tokenization. The model undergoes curriculum learning in three stages. In the first two stages, it is trained on Chinese data to support zero-shot long-context speech generation. In the third stage, we mix both Chinese and English conversational data to handle multilingual spontaneous generation tasks.

**Speech Detokenizer** For the speech detokenizer, we adopt a 0.8B-parameter, 10-layer Dit-style Transformer with a hidden size of 2048 and 16 attention heads. During training, the chunk size is dynamically set between 0.5 and 3 seconds to support flexible inference. For inference, we specifically use a chunk size of 3 seconds to achieve better quality. The backward ODE for each chunk is solved using 30 steps with the torchdyn toolkit [Poli et al.]. In addition, we adopt a 250M-parameter BigVGAN [Lee et al., 2022] to reconstruct waveforms from mel-spectrograms.

Table 1: The performance comparison on the Chinese podcast generation. **Bold** for the best result, and underline for the second-best result.

| Models | Subjective | | | | | Objective | |
|---|---|---|---|---|---|---|---|
| | Spontaneity (↑) | Coherence (↑) | Intelligibility (↑) | Quality (↑) | Similarity (↑) | SIM-O (↑) | WER (↓) |
| Cosyvoice2 | $3.68_{\pm 0.24}$ | $3.55_{\pm 0.28}$ | $4.18_{\pm 0.17}$ | $3.94_{\pm 0.26}$ | $\underline{3.94}_{\pm 0.21}$ | $\underline{0.85}$ | 2.40 |
| Concat Baseline | $\underline{3.94}_{\pm 0.26}$ | $\underline{3.98}_{\pm 0.27}$ | $\underline{4.38}_{\pm 0.15}$ | $\underline{4.00}_{\pm 0.31}$ | $\underline{3.94}_{\pm 0.21}$ | **0.86** | **1.90** |
| MoonCast | $\mathbf{4.33}_{\pm 0.17}$ | $\mathbf{4.26}_{\pm 0.21}$ | $\mathbf{4.43}_{\pm 0.12}$ | $\mathbf{4.13}_{\pm 0.16}$ | $\mathbf{4.19}_{\pm 0.15}$ | 0.77 | $\underline{2.15}$ |

Table 2: The performance comparison on the English podcast generation. **Bold** for the best result, and underline for the second-best result.

| Models | Subjective | | | | | Objective | |
|---|---|---|---|---|---|---|---|
| | Spontaneity (↑) | Coherence (↑) | Intelligibility (↑) | Quality (↑) | Similarity (↑) | SIM-O (↑) | WER (↓) |
| Cosyvoice2 | $\underline{3.86}_{\pm 0.24}$ | $\underline{3.88}_{\pm 0.24}$ | $\underline{4.46}_{\pm 0.14}$ | $\underline{4.25}_{\pm 0.18}$ | $\mathbf{4.40}_{\pm 0.10}$ | $\underline{0.73}$ | 2.77 |
| Concat Baseline | $3.73_{\pm 0.23}$ | $3.71_{\pm 0.21}$ | $3.93_{\pm 0.17}$ | $3.74_{\pm 0.19}$ | $3.96_{\pm 0.18}$ | **0.75** | $\underline{2.56}$ |
| MoonCast | $\mathbf{4.54}_{\pm 0.16}$ | $\mathbf{4.50}_{\pm 0.15}$ | $\mathbf{4.61}_{\pm 0.12}$ | $\mathbf{4.30}_{\pm 0.10}$ | $\underline{4.25}_{\pm 0.18}$ | 0.53 | **1.81** |

### 5.1.3 Evaluation Details

**Evaluation Dataset.** For podcast generation, we curate an evaluation dataset comprising two knowledge sources in PDF format and two in web URL format, encompassing domains such as computer science papers[4], economics papers[5], technology blogs[6], and news articles[7]. To verify the importance of spontaneous text, we select seven two-speaker Chinese podcasts, with speakers not present in the training data, totaling 125 turns, to assess the impact of scripted text on generation quality. For both datasets, we use 3-10 seconds of speech as the prompt for each speaker.

**Model Comparison.** We employ a concatenation baseline, whose text-to-semantic model is trained exclusively on single-speaker, single-turn data while other models remain the same. We also utilize Cosyvoice2 [Du et al., 2024b], a powerful open-sourced multi-lingual single-speaker zero-shot TTS model, as another baseline. For these two baselines, we first generate each dialogue turn individually in a zero-shot manner, and then concatenate these turns to form the complete podcast.

**Evaluation Metric.** We employ both subjective and objective metrics for a comprehensive evaluation. For the subjective evaluation, we involve ten evaluators to assess three specific aspects of the generated podcast: the entire audio, transitions between segments and individual segments. Specifically, we consider 1) spontaneity of the entire generated podcast and 2) coherence of transitions between segments. Additionally, for individual segments, we focus on three metrics: 3) intelligibility, 4) speech quality and 5) speaker similarity. For the objective evaluation, we employ SIM-O to assess speaker similarity and the Word Error Rate (WER) to evaluate robustness. In detail, we apply the pretrained WavLM-TDCNN[8] speaker embedding model to assess speaker cosine similarity between generated samples and the prompt speech. We average the SIM-O scores for each round according to the audio length. We use FunASR for Chinese speech transcription and NeMo ASR toolkit[9] for English. Note that we select NeMo ASR instead of Whisper because the Whisper model tends to suffer from hallucination issues.

### 5.2 Experimental Results

In this section, we first evaluate MoonCast by comparing it with existing baselines on the podcast generation task, thus confirming its superior performance. Subsequently, we empirically validate a key assumption of MoonCast: the spontaneity of the generated audio is significantly influenced by the spontaneity of the script text itself.

---

[4] https://arxiv.org/pdf/1706.03762

[5] https://gwern.net/doc/statistics/decision/1951-nash.pdf

[6] https://openai.com/index/hello-gpt-4o/

[7] https://www.nobelprize.org/prizes/physics/2024/press-release/

[8] https://huggingface.co/microsoft/wavlm-base-plus-sv

[9] https://huggingface.co/nvidia/parakeet-tdt-1.1b

Table 3: The influence of spontaneous scripts for podcast generation. **Bold** for the best result, and underline for the second-best result.

| Models | Subjective | | | | | Objective |
| --- | --- | --- | --- | --- | --- | --- |
| | Spontaneity (↑) | Coherence (↑) | Intelligibility (↑) | Quality (↑) | Similarity (↑) | SIM-O (↑) |
| GT | $4.73_{\pm 0.09}$ | $4.63_{\pm 0.08}$ | $4.57_{\pm 0.06}$ | $4.48_{\pm 0.10}$ | $4.57_{\pm 0.11}$ | 0.83 |
| GT script | $\mathbf{4.17}_{\pm 0.09}$ | $\underline{3.83}_{\pm 0.09}$ | $3.97_{\pm 0.08}$ | $\underline{3.97}_{\pm 0.11}$ | $\underline{4.00}_{\pm 0.11}$ | $\underline{0.68}$ |
| Written Script | $3.22_{\pm 0.09}$ | $3.53_{\pm 0.12}$ | $\underline{4.27}_{\pm 0.11}$ | $3.62_{\pm 0.12}$ | $3.67_{\pm 0.13}$ | $\underline{0.68}$ |
| Spontaneous Script | $\underline{4.03}_{\pm 0.10}$ | $\mathbf{4.00}_{\pm 0.11}$ | $\mathbf{4.53}_{\pm 0.11}$ | $\mathbf{4.03}_{\pm 0.12}$ | $\mathbf{4.03}_{\pm 0.12}$ | $\mathbf{0.72}$ |

### 5.2.1 Evaluation on Podcast Generation

To assess the efficacy of MoonCast, we evaluate podcast quality by comparing it with the two single-speraker baseline using the collected input knowledge sources. We report the evaluation results of the Chinese and English podcast generation in Table 1 and 2. We make the following observations: 1) MoonCast consistently surpasses the two concatenation baselines in terms of spontaneity, coherence, intelligibility and quality metrics for both Chinese and English podcast generation. Thus result demonstrates that the long-context two-speaker audio modeling captures contextual dependencies, thereby validating the effectiveness of our proposed method in generating high-quality results. 2) Despite the inherent systematic errors in the ASR model when handling proper nouns and filler words, MoonCast still achieves a WER of 2.15 for Chinese and 1.81 for English podcast generation, further demonstrating the robustness of the proposed system. 3) Furthermore, we observe a certain degree of discrepancy between the SIM-O and subjective similarity metrics, possibly because the single speaker embedding used by the SIM-O score may not fully capture all speaker characteristics, such as temporal features like prosody. Additionally, the relatively lower SIM-O score observed for English podcast generation may be attributed to our exclusive use of English conversational sources, which tend to be more prone to diarization errors.

### 5.2.2 Impact of Spontaneous Script

To investigate the impact of spontaneous script texts on the generation of spontaneous podcasts, we compare the generated speech using three types of input podcast scripts: 1) GT script: the ground-truth script obtained through our data preparation pipeline from the collected, unseen podcast speech. 2) Written script: We ask LLM to filter out spontaneous details from the GT script, resulting in the corresponding written version. 3) Spontaneous script: We ask LLM to reintroduce spontaneous details to the written script, resulting in the corresponding spontaneous version. To ensure a fair comparison, the same text-to-speech model is consistently applied across all script variations. The comparative results of the generated audio against the ground-truth audio are presented in Table 3. WER results are excluded due to recognition errors inherent in the ASR-derived transcripts. Our findings reveal several key insights: 1) The GT script, being the most spontaneous, achieves the highest spontaneity score. This score significantly decreases ($-0.95$ compared to the GT script) when spontaneous details are removed in the written script. Upon reintroducing these details in the spontaneous script, the score partially recovers, approaching that of the GT script ($-0.14$ compared to the GT script). This underscores the critical role of spontaneity in podcast text quality. 2) Generally, written scripts exhibit a larger training-inference mismatch compared to spontaneous scripts (both GT script and spontaneous script settings), often resulting in poorer performance. This is evidenced by a consistent performance deficit exceeding 0.3 across metrics of spontaneity, coherence, quality and similarity, further emphasizing the importance of spontaneous scripting. 3) The system consistently achieves commendable sim-o and intelligibility scores across various settings, demonstrating its robust capability for long-context generation. Nonetheless, we note that the intelligibility of the GT script is marginally affected by recognition inaccuracies in ASR transcripts. 4) Even with the use of the GT script, there remains a noticeable disparity in the quality of our generated audio compared to the GT audio, highlighting potential areas for future research. We hypothesize that several factors contribute to this performance gap: First, the data preparation pipeline may not be perfect, as the GT script still contains some recognition and diarization errors. Second, the GT audio contains rich and diverse spontaneous non-speech details, such as throat clearings.

## 6  Conclusion

Our work presents a novel solution for high-quality zero-shot podcast generation, addressing the key challenges of long speech duration and spontaneity that limit traditional text-to-speech systems. By adopting a long-context language model-based audio modeling approach and integrating a podcast generation module, MoonCast effectively synthesizes spontaneous, podcast-style speech from text-only sources using unseen speakers' voices. Experiments demonstrate that MoonCast outperforms existing baselines significantly in terms of contextual coherence, and spontaneity. This approach advances the state-of-the-art in text-to-speech for long and spontaneous dialogues, paving the way for more realistic and engaging podcast generation. We discuss our limitations and future work in Appendix G.

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

# A  Speech Semantic Codec Details

We adopt the semantic speech tokens [Borsos et al., 2022, Wang et al., 2024, Du et al., 2024a] which are discretized from the self-supervised learning (SSL) features due to the superiority of robustness [Wang et al., 2024]. Following the MaskGCT [Wang et al., 2024], we choose the 50-HZ SSL features and adopt the VQ-VAE approach to maintain the loss contained in the discrete semantic codes, thereby enhancing the reconstruction quality.

In detail, we train a VQ-VAE model to learn the discrete speech semantic representation by reconstructing the SSL features. For the SSL feature, we adopt the 17th layer of the pretrained W2v-BERT 2.0 [Schneider et al., 2019, Baevski et al., 2020, Chung et al., 2021, Barrault et al., 2023], and normalize it to mitigate the impact of varying scales across differnt feature dimensions. For the VQ-VAE model, we first encode the SSL feature $\mathbf{S}$ by an encoder and obtain $\mathbf{E}(\mathbf{S})$. Then we discrete the encoded feature by a VQ-VAE with a codebook and obtain the quantized speech feature $\hat{\mathbf{E}}(\mathbf{S})$. Finally, we apply a speech decoder to reconstruct the SSL feature $\mathbf{S}$ with a reconstruction loss. To enhance the codebook utilization and improve the reconstruction quality, we follow the design of improved VQ-GAN [Yu et al., 2021] and DAC [Kumar et al., 2023] to project the $\mathbf{E}(\mathbf{S})$ into an 8-dimension latent space.

# B  Data Preparation

Since the raw audio data contain artifacts such as background noise, overlapping speech, and reverberation, we first apply an automated data processing pipeline as described in [Yu et al., 2024, He et al., 2024]. Specifically, to improve speech quality, we use a band-split RNN speech enhancement model [Yu et al., 2022] to suppress background noise. Subsequently, speech diarization is performed using the Pyannotate toolkit[10] to segment the audio into distinct speaker segments. Finally, the Paraformer ASR model [Gao et al., 2022] from the FunASR toolkit[11] is utilized to generate pseudo-transcriptions for each segment. The DNSMOS toolkit[12] is also employed to evaluate speech quality. Additionally, to mitigate recognition errors introduced by the ASR system, a DNN-HMM-based forced alignment system is employed to align the pseudo-transcriptions with the speech audio, using a narrow beam size of 5. Only the speech segments with successful alignment are retained for subsequent processing. For curriculum learning training, we use all speech segments with a DNSMOS score greater than 2.6 to obtain single-speaker, single-turn speech. For long-context, two-speaker, multi-turn training data, we select two-speaker data based on our diarization results. Specifically, for conversational sources, we retain speech data involving exactly two speakers, with more than 10 conversational turns, and where the average duration of each turn is less than 30 seconds. This process results in a dataset comprising 300,000 hours from Chinese audiobook sources, 15,000 hours from Chinese conversational sources, and 200,000 hours from English conversational sources. Notably, to preserve the contextual information in the long speech data, we did not filter any segments based on DNSMOS scores or alignment results.

# C  Comparison with Dialogue Generation Baselines

We evaluate MoonCast against two state-of-the-art English dialogue generation baselines: Sesame[13] and Dia[14]. A notable limitation of these baselines is their maximum context length (2048 and 3072 tokens, respectively), which necessitates truncating long-range context during inference on podcast-length audio. As shown in Table 4, MoonCast achieves a substantially lower WER while maintaining a competitive SIM-O score, highlighting its superior ability to generate intelligible long-form audio.

---

[10]https://github.com/pyannote/pyannote-audio.git
[11]https://github.com/modelscope/FunASR
[12]https://github.com/microsoft/DNS-Challenge
[13]https://github.com/SesameAILabs/csm
[14]https://github.com/nari-labs/dia

Table 4: The performance comparison against dialogue generation baselines. **Bold** for the best result.

| Models | SIM-O (↑) | WER (↓) |
|---|---|---|
| Sesame | 0.53 | 2.71 |
| Dia | **0.54** | 3.10 |
| MoonCast | 0.53 | **1.81** |

# D  Ablation Study on Data Scale

To evaluate the effect of data scale, we train a model on a subset containing only 10% of the full dataset. The results, presented in Table 5, reveal a substantial drop in performance on English podcast generation across most subjective metrics, while only slightly impacting intelligibility. This underscores the critical role of large-scale data for high-quality podcast generation.

Table 5: Ablation study on data scale for English podcast generation. **Bold** for the best result.

| Models | Subjective | | | | | Objective | |
|---|---|---|---|---|---|---|---|
| | Spontaneity (↑) | Coherence (↑) | Intelligibility (↑) | Quality (↑) | Similarity (↑) | SIM-O (↑) | WER (↓) |
| 1/10 Data | $4.18_{\pm 0.14}$ | $4.23_{\pm 0.15}$ | $4.58_{\pm 0.13}$ | $4.18_{\pm 0.12}$ | $4.12_{\pm 0.15}$ | 0.50 | 1.94 |
| Full Data | $\mathbf{4.54}_{\pm 0.16}$ | $\mathbf{4.50}_{\pm 0.15}$ | $\mathbf{4.61}_{\pm 0.12}$ | $\mathbf{4.30}_{\pm 0.10}$ | $\mathbf{4.25}_{\pm 0.18}$ | **0.53** | **1.81** |

# E  Prompts

We choose 'Gemini 2.0 Pro Experimental 02-05'[15] for script generation because of its more conversational language style, natural dialogue design, and better topic coverage. We open-source LLM prompts to enhance reproducibility, covering brief generation and brief-to-script generation.

### E.1  English Prompt For Brief Generation

```
### Task Description
Please summarize the input document in plain text format according to
the following structure. The summary should be creative,
comprehensive, and include all interesting, uncommon, and valuable
viewpoints and information.

- **Text Requirements**:
    1. Directly output the result without any additional information.
    2. The summary should be in English. Retain a small number of
    proper nouns, names, and abbreviations in their original form
    (e.g., Chinese characters).
    3. Do not include any mathematical formulas.
    4. Do not alter any proper nouns, names, or abbreviations from
    the original text. Unless there is a common translation, do not
    translate proper nouns. Do not attempt to modify the meaning of
    proper nouns.
    5. **Intelligently convert numbers in abbreviations. For example,
    "a2b" should be interpreted as "a to b," not "a two b"; "a4b" as
    "a for b," not "a four b"; "v2" may represent "version two" or
    "second generation." Provide the original abbreviation and your
    suggested English translation.**

### Title and Author
- **Language Requirements**: English, formal written language.
```

---

[15]https://cloud.google.com/vertex-ai/generative-ai/docs/gemini-v2#2.0-pro

- **Content Requirements**: Provide the title and author of the document. Briefly summarize the theme of the document and the author's background. Ensure all important information is included without omission and sufficient context is retained.

### Abstract
- **Language Requirements**: English, formal written language.
- **Content Requirements**:
    1. What this document has done.
    2. Whether similar work has been done before.
    3. If similar work exists, why this document is still necessary.
    4. How this document specifically addresses the topic.
    5. How well this document achieves its goals.
- **Additional Requirements**: Include an additional paragraph to explain any terms, concepts, or methods that may confuse readers unfamiliar with the field. Ensure proper nouns are explained consistently with the original text, covering all potential points of confusion, including abbreviations and entity names.

### Main Themes and Concepts
- **Language Requirements**: English, formal written language.
- **Content Requirements**: Each theme and concept should be organized according to the 3W principle:
    - **What**: Clearly define the problem.
    - **Why**: Analyze the problem and identify its root causes.
    - **How**: Explain how the document addresses the problem.
- **Additional Requirements**:
    1. Ensure each theme and concept is comprehensive and includes all important details. Fully elaborate on the "What" and "Why" sections.
    2. Avoid technical details such as mathematical formulas in the "How" section. Use language that is easily understood by a general audience.
    3. Ensure themes and concepts do not overlap and maintain clear logic.
    4. Include an additional paragraph to explain any terms, concepts, or methods that may confuse readers unfamiliar with the field. Ensure proper nouns are explained consistently with the original text, covering all potential points of confusion, including abbreviations and entity names.

### Key Citations
- **Language Requirements**: English, formal written language.
- **Content Requirements**: Organize the content according to the following structure:
    1. **Argument**: State what needs to be proven.
    2. **Evidence**: Provide the material used to support the argument.
    3. **Reasoning**: Describe the process of using evidence to prove the argument.
- **Additional Requirements**:
    1. Ensure all evidence and reasoning are directly sourced from the original text without fabrication.
    2. Ensure citation content is complete and retains sufficient context without simplification. Avoid using mathematical formulas in citations.

```
    3. Include an additional paragraph to explain any terms, concepts,
    or methods that may confuse readers unfamiliar with the field.
    Ensure proper nouns are explained consistently with the original
    text, covering all potential points of confusion, including
    abbreviations and entity names.

### Conclusion
- **Language Requirements**: English, formal written language.
- **Content Requirements**: Highlight the most important and
impactful aspects of the document. Compared to the abstract, this
section should provide more detailed insights related to the main
themes and concepts. It may also include future directions for
improvement, current application scenarios, and existing challenges.
```

## E.2 Chinese Prompt For Brief Generation

```
### 任务说明
请按照以下结构总结输入文件，普通文本格式。总结应当有创造性，保证信息全
面，包含所有有趣、不常见、有价值的观点和信息。
- **文本要求**：
    1. 直接输出结果，不要包含任何额外信息。
    2. 总结文本用中文。允许少部分实体名词、专有名词、缩写等使用英文。
    3. 不要包含任何数学公式。
    4. 不要修改原文的任何实体名词、专有名词、缩写等。除非有常见译名，否
    则不要翻译实体名词。不要试图修改实体名词意思。
    5. **请智慧地将简写中的数字转化。如简称里"a2b''实际代表"a to b''，而
    不是"a二b"；简称里"a4b''实际代表"a for b''，而不是"a四b"；"v2''可能代
    表"version 二''，也可以进一步翻译成"第二代''。请提供原始简称，和你认
    为合适的中文翻译。**

### 标题和作者
- **语言要求**：中文，书面语。
- **内容要求**：提供文档的标题和作者。简要概括文档的主题和作者的背景。确
保包含所有重要信息，不要有遗漏，尽可能保留足够的信息。

### 摘要
- **语言要求**：中文，书面语。
- **内容要求**：
    1. 本文做了什么事情。
    2. 之前有没有别人做过这个事情。
    3. 如果有别人做过，那本文为什么还需要做。
    4. 本文具体怎么做的。
    5. 本文做的怎么样。
- **附加要求**：额外提供一个段落，解释本节中可能让听众困惑的术语、概念、
方法等，确保不了解领域的读者也能理解。专有名词的解释需贴合原文，覆盖所有
可能的困惑点，包括缩写名词、专有名词、实体名等。

### 主要主题和概念
- **语言要求**：中文，书面语。
- **内容要求**：每个主题概念需按照3W原则组织，包括：
    - **What**：界定问题，搞清楚问题是什么。
    - **Why**：分析问题，结构化分析问题本质原因是什么。
```

- – **How**：解决问题，文档如何解决问题。
- – **附加要求**：
    1．确保主题概念包含所有重要信息，不要有遗漏，主题概念需足够详细，充分阐述What和Why两个部分。
    2．How部分不要包含数学公式等技术细节。要用大众理解的语言充分概括。
    3．各主题概念间不要互相重叠，保证逻辑清晰。
    4．额外提供一个段落，解释本节中可能让听众困惑的术语、概念、方法等，确保不了解领域的读者也能理解。专有名词的解释需贴合原文，覆盖所有可能的困惑点，包括缩写名词、专有名词、实体名等。

### 重要引文
- – **语言要求**：中文，书面语。
- – **内容要求**：按照以下结构组织内容：
    1．**论点**：需要证明什么。
    2．**论据**：用于证明论点的材料。
    3．**论证**：运用论据证明论点的过程。
- – **附加要求**：
    1．论据和论证思路需严格来源于原文，不要进行任何虚构。
    2．确保引文内容充分，不要有遗漏，尽可能保留足够的信息，不要进行任何精简。引文避免使用数学公式。
    3．额外提供一个段落，解释本节中可能让听众困惑的术语、概念、方法等，确保不了解领域的读者也能理解。专有名词的解释需贴合原文，覆盖所有可能的困惑点，包括缩写名词、专有名词、实体名等。

### 总结
- – **语言要求**：中文，书面语。
- – **内容要求**：突出文档最重要、最吸引人眼球的部分。与摘要相比，需更结合主题概念的具体内容，对摘要进行补充。可包含未来改进方向、当前应用场景、当前存在问题等。

**E.3  English Prompts for Brief-to-Script Generation.**

```
## 1. Task Overview

Please generate a lively English podcast script based on the provided
English summary text and your knowledge of the topic. The script
should feature a dialogue between two speakers who take turns
speaking.  Output format should be JSON-parsable **list**. Each
speaker's turn is a **dictionary** containing "speaker" and "text"
fields. Example format: `[{{"speaker": "1", "text": "xxx"}}]`. The
"speaker" field indicates the speaker's identity (1 for host, 2 for
guest), and the "text" field is the spoken content. Output should
start directly with the JSON code block, without any extra
information.

## 2. Content and Structure
### (1) Text Content
- The summary text contains all important information, which needs to
be comprehensively selected and incorporated into the script.
- Present information through a dialogue between two speakers,
maintaining creativity and abstracting away unimportant details. For
example, listeners aren't concerned with specific test names, but
rather the task itself, the results, and the analysis.
```

### (2) Structure Design
- **Opening:** Introduce the topic and briefly describe the discussion content, without mentioning speaker names.
- **Key Theme Discussion:** Discuss important themes based on the summary text. Expand on the summary, don't just repeat it verbatim.
- **Closing:** Briefly recap the discussion highlights and offer an outlook on future or technological developments.

## 3. Language Style
### (1) Conversational Style
- The text should be as conversational as possible, aiming for a style similar to automatic speech recognition output. Include filler words such as 'um,' 'uh,' 'like,' 'you know,' 'so,' 'right?', and so on. Response words such as 'Yeah,' 'Right,' 'Okay,' and similar. Conversational expressions, repetitions, informal grammar, etc. Use short sentences. Avoid directly copying and pasting structured text from the summary text. Parentheses and other symbols not typically found in speech recognition transcripts should be avoided. Spaces within sentences indicate pauses. Be aware that there might be homophone errors, potentially due to accents. Questions should sound very conversational. Pay particular attention to incorporating conversational details, especially in questions. For example:
    [
    {{  "speaker": "1",
        "text": "Welcome back to the podcast, everyone. Today we're
        diving into, uh, something that's really changing everything
        around us, A I."
    }},
    {{  "speaker": "2",
        "text": "Yeah, A I is, like, everywhere now, isn't it?  It's
        kinda wild to think about."
    }},
    {{  "speaker": "1",
        "text": "Totally.  And we're seeing it in so many areas of
        daily life.  Like, even just recommending what to watch, or,
        you know, suggesting products online."
    }},
    {{  "speaker": "2",
        "text": "Mhm, exactly.  And it's not just online stuff,
        right? Think about smart homes, or even self-driving cars.
        It's getting pretty advanced."
    }},
    {{  "speaker": "1",
        "text": "Right, self-driving cars are still a bit futuristic
        for most of us, but, uh, even things like voice assistants on
        our phones, that's A I, isn't it?"
    }},
    {{  "speaker": "2",
        "text": "Definitely.  Siri, Alexa, Google Assistant, all
        powered by A I.  It's become so normal, we almost don't even
        think about it anymore."
    }},
    {{  "speaker": "1",
        "text": "Yeah, it's like, integrated into everything.  But is
        that a good thing, you think?  Like, are there downsides to
        all this A I in our lives?"
    }},

```
    {{  "speaker": "2",
        "text": "Well, that's the big question, isn't it?  On the one
        hand, it makes things so much more convenient, saves us time,
        maybe even makes things safer in some ways."
    }},
    {{  "speaker": "1",
        "text": "Safer how?"
    }},
    {{  "speaker": "2",
        "text": "Uh, well, like in healthcare, for example.  A I can
        help doctors diagnose diseases earlier, maybe even more
        accurately. That's a huge plus, right?"
    }},
    {{  "speaker": "1",
        "text": "Yeah, that's a really good point.  Medical
        applications are definitely exciting.  But what about the
        concerns, you know?  Like job displacement or privacy
        issues?"
    }},
    {{  "speaker": "2",
        "text": "Right, those are super valid concerns.  Job
        displacement is a big one. If A I can do more and more tasks,
        what happens to human workers?  And privacy,"
    }},
    {{  "speaker": "1",
        "text": "And privacy is huge, especially with all the data A
        I systems collect.  It's a lot to process."
    }},
    {{  "speaker": "2",
        "text": "Exactly.  So, it's not just sunshine and roses, is
        it?  We need to be mindful of the ethical implications and
        make sure it's used responsibly."
    }},
    {{  "speaker": "1",
        "text": "Definitely.  It's a powerful tool, but like any tool,
        it can be used for good or, you know, not so good.  It's up
        to us to guide its development, right?"
    }},
    {{  "speaker": "2",
        "text": "Absolutely.  And that's a conversation we all need
        to be part of, not just the tech people, but everyone."
    }}
    ]
```

### (2) Punctuation
- Use English punctuation marks. Avoid using other punctuation marks
beyond commas, periods, and question marks.  Exclamation points are
prohibited.  Ellipses ('...'), parentheses, quotation marks
(including ' ' " '' ") or dashes are prohibited, otherwise it will be
considered unqualified. do not use markdown syntax.  For
example,**bold** or *italic* text should be avoided.  Use plain text
only.
- If interrupted by the other person's response, the sentence should
end with a comma, not a period.

## 4. Information Organization and Logic
### (1) Referencing Issues

- Given that listeners won't have access to the summary text, any
references must provide sufficient context for comprehension.
- Avoid simply paraphrasing; instead, explain referenced content in
your own words.
- Explanations of technical terms should be creative and avoid simply
stating 'this means what?' You can use examples, metaphors, and so on
for explanations, but ensure you also clarify the rationale behind
the metaphor. Explanations can be provided in response to a question
from the other speaker, or you can offer explanations proactively.
Technical terms that are not mentioned don't need explanation.
Technical terms that are mentioned don't necessarily need immediate
explanation; they can be explained alongside other technical terms.
Technical terms in the summary text might differ slightly from the
surrounding text; you'll need to provide reasonable explanations
based on the context.
### (2) Information Density
- Ensure moderate information density, avoiding excessively high or
low density. The goal of appropriate information density is to enable
listeners without prior knowledge to quickly grasp the document's
purpose, rationale, and methodology.
- To prevent information overload, the script should avoid delving
into details like mathematical formulas, test setups, or specific
experimental metrics. Instead, it should use simple, generalized
language for descriptions.
- To avoid excessively low information density, ensure each topic is
discussed for at least 4 speaker turns, moving beyond simple keyword
listings. Discuss topics from multiple angles whenever possible,
going beyond the provided summary text. Given that the summary text
is highly generalized, the script should elaborate on it and discuss
further details. Feel free to use your knowledge to supplement
background information, provide examples, and so forth, to enhance
listener understanding.
- Techniques to increase information density:
    1. Incorporate memorable quotes. Add impactful,
    attention-grabbing sentences to the script, either original ones
    or quotes from other sources.
    2. Boost knowledge content.  Judiciously add knowledge points to
    the script to make listeners feel more informed and rewarded.
    3. Introduce novel information. Incorporate new concepts to spark
    listener curiosity, particularly information they're unaware of
    but would find valuable. This is crucial.
    4. Employ reverse thinking. Include information from diverse
    angles, challenging listeners' existing perspectives and
    presenting alternative viewpoints.
    5. Generate contrast and impact. The script can offer
    unconventional (yet plausible) descriptions of familiar concepts
    to create a contrast with listener expectations.  This contrast
    contributes to information density.
- Techniques to decrease information density:
    1. Use short sentences: Concise and easy to understand, making
    the narrative more compact. Do not have too much information in
    one sentence.
    2. Describe details: Vague and abstract information makes it
    difficult for listeners to build understanding, while more
    details create a sense of imagery and are easier to read.

3. Use more scenario-based descriptions: Scenarios are concrete and visual. Listeners can easily receive the conveyed information and be emotionally touched.
4. Talk more about facts: Talking about facts makes it more real, and readers can empathize more, thus lowering the information density of the copy.
5. Tell more stories: Tell your own stories, stories around you, and stories you've heard. Stories can bring listeners into the scene, making it easier to concentrate on listening.
6. Use more verbs and concrete nouns: Verbs and concrete nouns make it easier for listeners to visualize, while adjectives make complex copy harder to understand.
7. Avoid using mathematical formulas: Mathematical formulas are not conducive to public understanding.

## 5. Dialogue Design
### (1) Speaker Roles
- The script includes a host and a guest. Speaker 1 is the host, responsible for opening and closing the show, skilled at using questions to control the pace of the conversation, and using vivid examples to make knowledge less dry. Speaker 2 is the guest, primarily responsible for introducing the document content, has amazing knowledge reserves in the field, and is good at organizing language in a structured and easy-to-understand way.
- Both speakers are enthusiastic and cheerful, like to combine personal stories or examples for discussion, and bring a direct experience to listeners. They are happy to discuss digressive stories.
- The two speakers actively interact and frequently use interruption words such as "um" to indicate agreement with each other. Response words need to be inserted into the dialogue according to the timing. Sentences before being interrupted end with a comma, not a period.
- Ensure consistent speaker roles. Do not have the host introduce technical details, or have the guest guide the host to discuss topics.
- The host gradually increases their understanding of the field based on the guest's answers. However, the host may not understand immediately or completely correctly. The host can express misunderstanding or raise some questions that ordinary people might have. In this case, the guest will further explain in more accessible language, or specifically answer common questions or misunderstandings. This kind of interaction is more realistic and easier for listeners to understand than always correct hosts and guests.
### (2) Topic Order Arrangement
- The host will arrange the topics according to the summary text and ensure logical connections between topics, such as transitioning from overall to details, from details to overall, from cause to effect, from technology to application, etc.
- The host will guide the pace of the conversation and discuss topics in the order of the summary text. Guests should not interfere with topic transitions.
### (3) Knowledge Rate
- The knowledge rate in the script needs to be reasonable. Do not introduce a large amount of knowledge too quickly in a short period of time. Knowledge

## 6. Other Requirements
### (1) English Numbers and Foreign Words
  1. The script will be used for English podcast content recording.
  Please ensure most numbers and foreign words are rendered naturally
  in English to facilitate correct pronunciation.
  2. Please intelligently determine the correct pronunciation
  according to the context. For example, "2021" if expressing a year,
  should be converted to "two thousand and twenty-one" or "twenty
  twenty-one". But if expressing a number, it should be "two thousand
  and twenty-one". For some uncommon English abbreviations, if the
  pronunciation needs to be read letter by letter according to the
  context, you must ensure that there a space between each letter,
  such as "AI" adding a space as "A I", to avoid the model
  misinterpreting it as a word. For example, "API" should be rendered
  as "A P I".
  3. Small amount of Chinese is allowed, especially for nouns, if it
  fits naturally within the conversational English context.
### (2) Script Length
  1. Please ensure that the total length of the 'text' values does
  not exceed 3,000 words and the number of speaker turns is kept
  within 60, otherwise it will be unqualified. Please choose
  technical details and topic concepts to discuss. Do not shorten the
  depth of discussion on each topic for the sake of word limit, do
  not be limited to the summary text, and give full play to your
  knowledge.

INPUT: {BRIEF}

## Re-emphasize:
Speaker 1 is the host, and Speaker 2 is the guest. Neither speaker
has a name. The script text only uses commas, periods, and question
marks. Use English punctuation marks. Avoid using other punctuation
marks beyond commas, periods, and question marks. Exclamation points
are prohibited.  Ellipses ('...'), parentheses, quotation marks
(including ' ' " '' ") or dashes are prohibited, otherwise it will be
considered unqualified.  Please prioritize in-depth discussion for
each topic. Don't limit yourself to the summary text, instead, use
your knowledge to expand upon the topics, providing background
information and illustrative examples to enhance listener
understanding.
Ensure that numbers and foreign words are rendered naturally in
English for accurate pronunciation during recording. In technical
contexts, English abbreviations sometimes use numerical digits in
place of words (e.g., "a2b" for "a to b," "a4b" for "a for b").
Please translate these abbreviations into appropriate English phrases
based on the context. While the script is primarily in English, a
small amount of Chinese, especially for nouns, is acceptable if it
integrates naturally into the conversational flow.

OUTPUT:

**E.4 Chinese Prompts for Brief-to-Script Generation.**

## 一、任务概述
请根据提供的总结文本，和你对这方面了解的知识，生成一个生动的中文播客文字剧本。 剧本包含两位说话人交替发言。输出格式为 JSON 可解析的**列表**。列表里每条发言是一个**字典**，包含"speaker"和"text"字段。示例格式：`[{{"speaker": "1", "text": "xxx"}}]`。"speaker"字段是说话人身份（1表示主持人，2表示嘉宾），"text"字段是具体发言内容。输出直接从json的代码块开始，不要包含任何额外的信息。

## 二、内容与结构要求
### （一）文本内容
- 总结性文本包含所有重要信息，需全面挑选并纳入剧本。
- 通过两位说话人的对话形式展示信息，保持创作性，适当抽象不重要的细节。例如，听众不关心具体的测试名称，而关心测试的任务，结果和分析。
### （二）结构设计
- **开场白**：引入主题，简要介绍讨论内容，不提及说话人姓名。
- **关键主题讨论**：逐字阅读总结文本，讨论重要主题。
- **结束语**：简洁总结讨论亮点，并对未来或技术发展进行展望。

## 三、语言风格
- 文本要尽量口语化，接近自动语音识别的结果，包含填充词
如"嗯"、"啊"、"呃","呢","这个","其实","就是","然后"等，响应词
如"嗯。"或"是。"等。多用口语化的表达方式，允许重复，语法可以不那么正式。避免直接照搬总结文本里的书面语。不要用括号或语音识别通常不会出现的符号。句中的空格代表短停顿，逗号表示稍长停顿，句号表示长停顿。可能存在因口音带来的同音识别错误。提问需要非常口语化。总之，就是要像平时聊天一样自然。示例如下:

```
[
{{  "speaker": "0",
    "text": "欢迎收听今天的播客。那我们这一集是要聊什么东西呢? ",
}},
{{  "speaker": "1",
    "text": "我们要聊星座。",
}},
{{  "speaker": "0",
    "text": "星座嘛，就是，他是一个好跟新的朋友认识的时候一个聊天的
    话题。",
}},
{{  "speaker": "1",
    "text": "没错，现我觉得在现在已经从你好，变成了诶，请问你的星座
    是什么呢? 。",
}},
{{  "speaker": "0",
    "text": "对，那我天枰座。",
}},
{{  "speaker": "1",
    "text": "那，我是摩羯座。",
}},
{{  "speaker": "0",
    "text": "摩羯座，那你会觉得就是星座，是一个可以相信的东西吗? ",
}},
{{  "speaker": "1",
```

```
           "text": "我本人其实不太相信星座诶，在一开始的时候。我就跟大部分
           不相信星座的一样，觉得，呃，你总能把人就分成十二种，然后呢就它讲
           的就是对的。",
       }},
       {{  "speaker": "0",
           "text": "啊，所以就是，会觉得说把星座就是单纯把人分成十二种事件
           很粗略，不太有什么科学根据的事情。",
       }},
       {{  "speaker": "1",
           "text": "嗯，对，会这样觉得。",
       }},
       {{  "speaker": "0",
           "text": "嗯。",
       }},
       {{  "speaker": "1",
           "text": "会无法理解，到底是，那这一开始定出这十二种人格的是谁
           啊？",
       }},
       {{  "speaker": "0",
           "text": "对，就是凭什么他可以决定，我们就是这十二种人格。",
       }},
       {{  "speaker": "1",
           "text": "嗯？",
       }},
       {{  "speaker": "0",
           "text": "为什么不是十三、十四或者更多的种类。",
       }},
       {{  "speaker": "1",
           "text": "对，没有错。",
       }},
       {{  "speaker": "0",
           "text": "对。那，所以你会觉得说那种就是什么星座的心理分析是完全
           不可信的，还是其实也会很常去看一下，呃，类似的这种星座测验。",
       }},
       {{  "speaker": "1",
           "text": "其实我刚说一开始不相信啊，我真的是到后期比较相信。然后
           后期会开始相信的是因为，呃，要去找一些我自己没有办法有方法去理解
           的人，因为认识那样子的人，他就是暧昧对象，必须要了解他到底是怎样
           的人，可是没有其他的依据的时候呢，我就偷偷开始看起了星座，然后就
           偷偷我觉得，好像讲得有那么一点准，然后就会开始看了。",
       }},
       {{  "speaker": "0",
           "text": "哦，所以感觉有点像是说在从，星座的这种描述测验中去找
           说，你想要从这个东西，去对那个人有更深一层的了解的感觉。",
       }},
       {{  "speaker": "1",
           "text": "对，而且通常他会讲到一两个你好你觉得好像是那样子的点，
           那你就会想要看更多，然后就好像就跟着就开始相信这个东西了。",
       }},
       {{  "speaker": "0",
           "text": "哦，嗯，诶，所以你是什么什么星座的？",
       }},
       {{  "speaker": "1",
           "text": "就我刚刚说我是摩羯座啊。",
       }}
```

```
    ]
```

### （二）标点符号
- 使用中文标点符号，避免英文标点。
- 剧本文本只使用逗号，句号和问号。禁止使用叹号。禁止使用省略号（'...'）、括号、引号（包括''""'"）或波折号，否则视为不合格。
- 如果被对方的响应词等打断，本句句末是逗号，而不是句号。

## 四、信息组织与逻辑
### （一）引用问题
- 由于听众看不到总结性文本，引用需确保上下文完整，确保听众能理解。
- 避免直接复述，需用自己的话解释引用内容。
- 总结文本里提供了对专业术语的解释。你需要保证你剧本里的专业术语尽可能被充分解释。专业术语的解释请具有创意，不要简单地创作成"这个是什么意思"这样的句子。可以通过举例、比喻等方式进行解释，但需要进一步说明比喻的合理性。可以由对方提问后进行解释，也可以自行解释。没有提到的专业名词不需要解释。提到的专业名词不一定要立即进行解释，可以和别的专业名词一起解释。总结文本中的专业术语可能与文字内容存在差异，你需要根据上下文合理解释。
### （二）信息密度
- 确保信息密度适中，避免过高或过低。适当的信息密度希望让没有相关背景知识的听众，快速理解文档里在做什么，为什么这么做，以及如何做。
- 为了避免信息密度过高，剧本不能讨论数学公式、测试设置、实验指标等细节，而应该用简单概括性语言描述。
- 为了避免信息密度过低，剧本每个主题需不少于4次发言，避免停留于关键词的简单罗列。会从尽可能从不同角度讨论，不局限于提供的总结文本。总结文本高度概括，剧本应当将其展开，讨论更多细节。你可以利用自己知识，补充背景知识，举例说明等方式，让听众更好地理解。
- 提高信息密度技巧：
    1．嵌入金句。在剧本中加入令人印象深刻，眼前一亮的句子，可以是自己创作，也可以是引用他人。
    2．增加知识点： 在剧本中适当增加知识点，能让听众听完更有收获。
    3．引入新信息：剧本中加入新的概念，引起用户好奇，特别是听众不知道但想知道的信息，这种非常重要。
    4．逆向思维： 加入不同角度的信息，打破用户熟悉的视角，提出不一样的观点。
    5．制造反差冲击： 剧本可以对用户熟知的认知进行非常规（出乎意料）但合理的描述，形成与他预期的反差，这种反差是信息密度。
- 降低信息密度技巧：
    1．使用短句：简洁明了，易于理解，让叙述更紧凑。不要一句话里有过多的信息。
    2．描述细节：模糊不清，抽象的信息难以让听众建立认知，而细节越多，越能有画面感，容易阅读
    3．多进行场景化塑造： 场景是具象的，有画面的。 听众能轻松接收传达的信息，还能让人触景生情。
    4．多讲事实：讲事实才能更显真实，读的人才能更感同身受，这样文案信息密度更低。
    5．多讲故事：讲自己的故事，讲身边的故事，讲听说的故事，故事能把听众带入场景，更利于聚精会神地收听。
    6．多用动词和具体名词：动词和具体的名词更容易让听众浮现画面，而形容词会让复杂的文案更难理解。
    7．避免使用数学公式： 数学公式不利于大众理解。

## 五、对话设计
### （一）说话人角色
- 剧本中包含主持人和嘉宾。其中说话人1是主持人，负责节目开场和结束，擅长利用提问控制对话节奏，用生动的例子让知识不枯燥。说话人2是嘉宾，是主要负责文档内容的介绍，对该领域有惊人的知识储备，擅长有条理地语言组织，通俗地讲解内容。
- 两位说话人热情开朗，喜欢结合个人故事或者实例进行讨论，给听众带来直观的体验。大家乐于讨论离题的故事。
- 两位说话人积极互动，会经常用"嗯"等打断词表示对对方的认同。需要将响应词按照时间点插入对话。被打断前的句子句末用逗号，而不是句号。
- 保证说话人角色统一，不要出现主持人介绍技术细节，或者引导主持人讨论主题等情况。
- 主持人根据嘉宾的回答，逐步增加对该领域的认知。但主持人不一定立刻能理解，也不一定理解地完全正确。主持人可以表达不理解或者提出一些常人可能会存在的疑问。这种情况下，嘉宾会进一步用更通俗的语言解释，或者针对性地解答常人常有的疑问或者误解。这种互动相比于永远正确的主持人和嘉宾更加真实，也更利于观众地理解。

### （二）主题顺序安排
- 主持人会根据总结性文本，将主题排列，并保证主题间有逻辑关联，如从整体过渡到细节，从细节过渡到整体，从原因过渡到结果，从技术过渡到应用等。
- 主持人会引导对话节奏，按照总结性文本的主题顺序进行讨论。嘉宾不应该干扰主题过渡。

### （三）知识速率
- 剧本中知识速率需要合理，不能短时间过快引入大量知识。知识不能突然增加，要逐渐引入，确保听众能够理解。
- 听众视角：充分考虑听众感受，从听众视角进行剧本创作。必须保证剧本不包含详细数学公式，而应该用通俗的语言介绍。确保剧本内容易懂，不要过于专业化。
- 无论是与主题相关的信息，还是离题的故事，都要按照你的知识进行充分地讨论，切忌简单地提一句而没有展开。要保证剧本足够真实，符合日常对话的逻辑，保证说话人间足够的尊重，不敷衍，不随意打断。

## 六、其他要求
### （一）外语数字：
  1. 剧本将用于中文播客内容的录制。请保证大部分外语和数字转换为中文，以便于模型能正确识别读音。
  2. 请根据上下文，智慧地判断正确的读音。例如，"2021''如果表达年份，应当转换为"二零二一''。但如果表示数字，应当转换为"两千零二十一''。一些英文简称里常用数字代表英文单词，比如"a2b''代表"a to b''，"a4b''代表"a for b''，请保证不要简单转换为中文数字，而是根据上下文，将其翻译成合适的中文。
  3. 对于一些不常见的英文简写，如果根据上下文判断读音需要逐个字母阅读，则须保证每个字母间留有空格，如"AI''添加空格为"A I''，以避免模型误认为是一个单词。除非实体名字有常见的中文翻译，否则不要翻译实体名字。
### （二）剧本长度
  1. 请控制"text"值的文本总长度不超过3000字符，且不超过60个发言，否则不合格。请选择技术细节，主题概念进行讨论。不要为了字数限制缩短每个话题讨论的深度，不要局限于总结文本，充分发挥你的知识。

INPUT: {BRIEF}

再次强调：

```
说话人1是主持人，说话人2是嘉宾。说话人和嘉宾没有姓名。剧本文本只使用逗
号，句号和问号。禁止使用叹号。禁止使用省略号（'...'）、括号、引号（包
括'，"，'"）或波折号，否则视为不合格。请优先保证每个话题讨论的深度，不要局
限于总结文本，利用你的知识，补充背景知识，举例说明等方式，让听众更好地理
解。
请保证大部分外语和数字转换为中文，以便于模型能正确识别读音。在技术文档
里，英文简称常用数字代表英文单词，比如"a2b"代表"a to b"，"a4b"代表"a for
b"，请保证不要简单转换为中文数字，而是根据上下文，将其翻译成合适的中文。

OUTPUT:
```

# F    Discussions

## F.1    Phoneme vs. BPE for Text Representation

Traditional TTS systems tend to use phonemes as the text representation to enhance intelligibility, but this pronunciation-based approach strips away semantic information needed for long-form, multi-speaker scenarios, hindering natural speaker transitions, emotion, and prosody. In contrast, we opt for BPE, which preserves semantic content and aligns with the text representation used in LLMs, thereby enabling more straightforward future integration. Empirically, BPE maintains intelligibility while improving prosody and spontaneously generating paralinguistic phenomena like laughter based on context.

## F.2    Hallucination Issues

We observe that hallucinations sometimes occur in the generated speech, that is, the synthesized output may confuse the identity of speakers, leading to incorrect attributions of utterances. These issues stem from the interplay of three main factors: First, the semantic tokens retain some timbre information, enabling reconstructed speech to deviate from the prompt's timbre. Second, the data pipeline may introduce errors, such as speaker identification errors or diarization errors, especially in distinguishing rapid transitions between speakers. Third, ambiguous text interpretations complicate the generation fprocess. For example, the sentence 'Today, we're discussing climate change um and its impact on global biodiversity.' can be interpreted in several ways. It might be understood as a single speaker using 'um' as a filler, such as: 'Host: Today, we're discussing climate change, um, and its impact on global biodiversity.' Alternatively, it could be interpreted as a dialogue between two speakers, such as: 'Host: Today, we're discussing climate change. Guest: Um. Host: And its impact on global biodiversity.' Therefore, the model struggles to determine whether the 'um' is a filler word from the same speaker, or a response word from another speaker, even with adequate semantic understanding. Additionally, we find that the trade-off between increasing sampling diversity (i.e., increasing temperature, top-k and top-p values) to enhance spontaneity and the consequent aggravation of hallucinations restricts the model's ability to achieve higher levels of spontaneity.

# G    Limitations and Future Works

Despite our proposed system has achieved great progress, we still have the following limitations:

**Language Coverage.** The current system is limited to Chinese and English. Future work should focus on expanding language coverage to support multiple languages to enhance the system's applicability in diverse linguistic contexts.

**Multi-Speaker.**: The system is currently designed for two-person interactions. Extending it to handle multi-person conversations is an important direction for future development to accommodate more complex and dynamic conversational scenarios.

**Data Quality.**: The data pipeline currently generates data that may contain errors in ASR and diarization. In addition, current data preparation pipeline filters out the overlapped part. To address

these challenges, future work should prioritize the use of high-quality, human-annotated spontaneous speech for fine-tuning.

## H   Broader Impacts

Given our model's ability to generate speech with high fidelity to the original speaker's voice, there is a risk of improper application, including deceptive voice recognition or mimicking an individual's speech. To mitigate potential abuse, it is crucial to devise a reliable method for detecting synthetic speech and implement a mechanism for flagging suspected malicious use.

