# OpenReview forum: "MoonCast: High-Quality Zero-Shot Podcast Generation"
_NeurIPS.cc/2025/Conference — NeurIPS 2025 poster_

### Official Review · Reviewer_biCE · 2025-06-09

**Clarity:** 2
**Significance:** 3
**Originality:** 4
**Rating:** 4
**Confidence:** 4

**Summary:**

This paper introduces a pipeline for podcast generation, comprising two main modules: 1) a script generation module that transforms text sources (e.g., news website URLs) into podcast scripts, and 2) an audio modeling module that converts these scripts into audio podcasts. The authors argue that existing text-to-speech (TTS) systems often struggle with generating long, complex podcasts that include spontaneous speech. To address these challenges, they propose pre-training a long-context audio model to enhance contextual coherence and generate spontaneous speech based on automatic speech recognition (ASR) transcripts. During training, the audio model is conditioned on prompt text, speaker prompts, and podcast scripts to model audio tokens. The training process employs a curriculum learning strategy, beginning with single-speaker, single-turn TTS tasks and progressing to long-context, multi-speaker audio sequence modeling. Experimental results demonstrate that the proposed pipeline outperforms baseline methods, such as CosyVoice2, in human evaluations of intelligibility and speech quality.

**Questions:**

See weaknesses.

**Ethical Concerns:**

["NO or VERY MINOR ethics concerns only"]

**Final Justification:**

Most of my concerns have been addressed by the author and I've adjust my score accordingly

**Limitations:**

Yes

**Quality:**

3

**Strengths And Weaknesses:**

Strengths:
- The framework is novel in addressing the challenges of generating long, complex podcasts that incorporate spontaneous speech.
- A long-context TTS model, pre-trained with half a million hours of audio, is released. This model is capable of generating podcast audio of up to approximately 13 minutes.

Weaknesses:
- The details of the large language model (LLM) used in the script generation module are not thoroughly discussed in the main paper (hidden in a small section in appendix).
- Several details are missing in the evaluation:
   - The number of samples in the evaluation set and the distribution across different domains.
   - The definition and rating guidelines for all subjective metrics.
   - The evaluation process lacks clarity: How many raters were hired? How many reviews were conducted per evaluation sample?
- Evaluation comprehensiveness: podcast generation latency and the factualness of generated podcast is not discussed in the paper.

---

> ### Author Rebuttal · Authors · 2025-07-31
>
> First of all, we want to thank the reviewer for your careful reading and providing a lot of constructive comments! Below we address the concerns mentioned in the review
>
> **Weakness-1. The details of the large language model (LLM) used in the script generation module are not thoroughly discussed in the main paper (hidden in a small section in appendix).**
>
> A1. Thank you for the feedback. We would like to clarify that these details are discussed in detail in Section 4.2 of the main paper. We also open-source the LLM prompts for script generation in Appendix C.
> To further elaborate, our LLM-powered module operates in three key stages:
> 1. Content Analysis: The module first processes user-provided input, such as a web URL or PDF, and employs an LLM to recognize its content.
> 2. Briefing Document Generation: We find empirically that generating a script directly from raw content can lead to vague outputs and a significant loss of information. To address this, we introduce a crucial intermediate step: generating a structured briefing document. This document synthesizes the source material into five key components: a title, an abstract, main topics, key citations, and a conclusion. It also includes explanations for technical terms to ensure clarity.
> 3. Script Generation: Finally, using this briefing document as a high-quality guide, the LLM generates the final podcast script. We guide this process by specifying the desired structure (e.g., engaging openings and closings), format (a two-speaker JSON), and content and style. Critically, to infuse the text with spontaneity, we explicitly prompt the LLM to incorporate elements like filler words (e.g., ‘um’, ‘uh’), response words (e.g., ‘yeah’, ‘right’), informal grammar, and repetitions. This encourages the output to stylistically match our ASR training data. To reinforce this, the prompt also includes an example ASR transcript as a demonstration, further guiding the LLM's output.
>
>
> **Weakness-2. Several details are missing in the evaluation:**
> - **The number of samples in the evaluation set and the distribution across different domains.**
> - **The definition and rating guidelines for all subjective metrics.**
> - **The evaluation process lacks clarity: How many raters were hired? How many reviews were conducted per evaluation sample?**
>
> A2.  Thank you for the detailed feedback regarding our evaluation methodology. We are happy to provide the following clarifications.
>
> Details on our evaluation setup are provided in Section 5.1.3. Specifically, our evaluation set consists of four samples derived from diverse knowledge sources: a computer science paper (PDF), an economics paper (PDF), a technology blog (URL), and a news article (URL). For the evaluation presented in the paper, we hire 10 raters. Each generated audio sample from every system is assessed by each rater, resulting in a total of 10 ratings per sample.
> To address your request for the rating guidelines, here are the specific instructions and questions provided to the evaluators for each subjective metric:
> - Spontaneity: Assesses how spontaneous the audio sounds. This focuses on the presence of common spoken elements (e.g., filler words such as 'um,' 'uh,' 'like,' 'you know,' 'so,' 'right?', and response words such as 'Yeah,' 'Right,' 'Okay') as well as a fluid and casual flow. Rated on a 1-5 scale with 0.5-point increments, where 5 is best.
> - Coherence: Evaluates the overall coherence of the audio. This includes the naturalness of transitions between speakers and whether the audio is globally consistent. Rated on a 1-5 scale with 0.5-point increments, where 5 is best.
> - Intelligibility: Measures whether the speech is clearly articulated and consistent with the provided text. Rated on a 1-5 scale with 0.5-point increments, where 5 is best.
> - Quality: Assesses the overall quality of the audio. Rated on a 1-5 scale with 0.5-point increments, where 5 is best.
> - Similarity: Determines if the speaker's voice is consistent with the voice in the provided reference audio. Rated on a 1-5 scale with 0.5-point increments, where 5 is best.
>
> Furthermore, we conduct a new, more comprehensive subjective evaluation on English podcast generation during the rebuttal period. This new round involves an increased pool of 20 raters and an expanded set of 10 distinct knowledge sources.  In this round, the audio is segmented by speaking turns, and each turn is rated individually. The results are presented below:
>
>
> | | Spontaneity(↑) | Coherence(↑) | Intelligibility(↑) | Quality(↑) | Similarity(↑) | SIM-O(↑) | WER(↓) |
> |---|:---:|:---:|:---:|:---:|:---:|:---:|:---:|
> |Cosyvoice2| 3.75±0.04 | 3.80±0.04 | 4.55±0.03 | 4.35±0.04 | **4.54±0.02** | 0.76 | 2.53 |
> |Concat Baseline| 3.62±0.03 | 3.65±0.04 | 4.12±0.03 | 3.65±0.03 | 4.01±0.03 | **0.78** | 2.36 |
> |Our System| **4.60±0.02** | **4.53±0.03** | **4.61±0.02** | **4.38±0.02** | 4.22±0.04 | 0.56 | **1.69** |
>
> **Weakness-3. Evaluation comprehensiveness: podcast generation latency and the factualness of generated podcast is not discussed in the paper.**
>
> A3. Thank you for raising these important points. Considering the term "podcast" here can refer to either the generated text script or the final synthesized audio, so we will address the latency and factualness for both aspects.
>
> If "podcast" refers to the text script:
>
> - Latency: The script generation latency is primarily determined by the external LLM used. In our implementation with the Gemini API, the process of converting a knowledge source into a full podcast script takes approximately 5 minutes on average.
> - Factualness: The factualness of the generated text is tied to the broader challenge of hallucination in LLMs. While this remains an open research problem without standardized methods for quantitative evaluation, we can report that in our extensive testing, we have observed very few instances of factual inaccuracies in the generated scripts.
>
> If "podcast" refers to the synthesized audio:
> - Latency: For audio synthesis, latency is best measured by the Real-Time Factor (RTF). Our audio modeling module achieves an average RTF of 0.9 on a single Nvidia H20 GPU. In practice, tensor parallelism can be employed to significantly reduce the RTF.
> - Factualness: The "factualness" of the audio is its fidelity to the input script, which is measured by Word Error Rate (WER). We would like to clarify that we do report this objective metric in the paper, and the low WER demonstrates that our model faithfully synthesizes the content from the provided script.

---

> ### Author Response · Authors · 2025-08-04
>
> With the discussion deadline approaching, we want to gently check in and see if our rebuttal was helpful in addressing your concerns. We are ready to provide any further clarification.

---

> ### Author Response · Authors · 2025-08-06
>
> Thank you for acknowledging our rebuttal. As there are no accompanying comments, we want to politely check in and ensure we have fully addressed your initial concerns. If any points remain unclear, we would be grateful for the opportunity to provide further clarification before the discussion period ends.

---

> > ### Comment · Reviewer_biCE · 2025-08-06
> >
> > Thanks for your responses, I've adjust my score accordingly.

---

> > > ### Author Response · Authors · 2025-08-07
> > >
> > > Thank you for the positive update. We're glad we could resolve your concerns. We truly appreciate your time and reconsideration

---

### Official Review · Reviewer_DLz7 · 2025-06-15

**Clarity:** 3
**Significance:** 2
**Originality:** 2
**Rating:** 4
**Confidence:** 4

**Summary:**

This paper proposes a pipeline for generating podcasts, specifically focusing on conversations between two speakers. The pipeline is designed to generate a podcast script from given content, convert the script into a speech code, and then reconstruct the code into speech. It integrates various components such as zero-shot generation, curriculum learning, and chunk-wise generation to form a cohesive system. The authors present their main experimental results in the main text.

**Questions:**

The questions I had were addressed in the Weaknesses.

**Ethical Concerns:**

["NO or VERY MINOR ethics concerns only"]

**Final Justification:**

I agree with the overall novelty of the framework proposed by the authors. However, I had pointed out the insufficient proof of effectiveness for several of its constituent elements and requested relevant experiments. The authors have addressed this, and therefore, I am changing my score to positive.

**Limitations:**

The limitations of the work are well documented in the Appendix.

**Paper Formatting Concerns:**

There do not appear to be any particular issues.

**Quality:**

2

**Strengths And Weaknesses:**

### **Strengths**

**(S1)** To achieve their objectives, the authors propose and incorporate several components, combining original ideas with existing methods. They aim to demonstrate the effectiveness of their model and contribute to the research community through a demo and a promise to release the code, although it has not yet been made publicly available.

**(S2)** The description of their methodology is written in a clear and accessible manner, and the various figures included throughout the paper effectively support the reader’s understanding.

### **Weaknesses**

**(W1)** To achieve their objectives, the authors incorporate various elements, including the adoption of techniques such as chunk-wise flow matching models, also employed in other recent studies (e.g., CosyVoice 2, 2024.12), and the use of a large language model (Gemini 2.0 Pro) for podcast script generation. They also propose several techniques, including sequence design and curriculum learning. While these proposed components are promising, it is unfortunate that their individual contributions and effectiveness are not clearly demonstrated through experiments, either in the main paper or the Appendix.

- **(W1-1)** What kind of performance degradation occurs if the model is trained directly without the proposed curriculum learning? Demonstrating such differences could help highlight the effectiveness of the components proposed in the paper.

- **(W1-2)** Similarly, I am curious about the advantages of the interleaving approach you adopted compared to the turn-based interleaving method mentioned in line 163, particularly in terms of performance, sample quality, or perceptual (auditory) improvements.

---

> ### Author Rebuttal · Authors · 2025-07-31
>
> First of all, we want to thank the reviewer for your careful reading and providing a lot of constructive comments! Below we address the concerns mentioned in the review.
>
> **Weakness-1. To achieve their objectives, the authors incorporate various elements, including the adoption of techniques such as chunk-wise flow matching models, also employed in other recent studies (e.g., CosyVoice 2, 2024.12), and the use of a large language model (Gemini 2.0 Pro) for podcast script generation. They also propose several techniques, including sequence design and curriculum learning. While these proposed components are promising, it is unfortunate that their individual contributions and effectiveness are not clearly demonstrated through experiments, either in the main paper or the Appendix.**
>
> A1. Thank you for your insightful question. First, we wish to clarify our contributions, as our paper is one of the first academic works to systematically address long-form, multi-speaker podcast generation. Our academic contributions include 1) proposing a robust pipeline for this task with novel script and audio generation modules; 2)identifying a key blind spot in the current TTS field, that is, the overlooked impact of text quality on speech spontaneity; and 3) introducing a chunk-wise causal mask to accelerate our speech detokenizer's training. On the engineering side, we 1) validate the powerful potential of scaling in this new TTS research direction and 2) plan to open-source our entire pipeline, including LLM prompts and audio module, to ensure reproducibility and establish a foundational baseline for this emerging research area.
>
> Regarding component-wise evaluation, we conduct a crucial ablation study in Section 5.2.2 (Table 3), which validates our most central insight about the impact of  spontaneous text script. We acknowledge that we do not perform an exhaustive ablation of every single component, as our overarching strategy was to present a simple enough yet highly effective system to demonstrate the immense potential of this research direction as a whole.
>
> To further prove our system's holistic advantage, we conduct a new comparison against current SOTA spontaneous conversational TTS models, including Sesame and Dia,  during this rebuttal period. A critical challenge in this comparison is that both Sesame and Dia have significant context length constraints (2048 and 3072 tokens, respectively), forcing them to discard parts of the long-range context during inference for podcast-length audio. As shown in the results below, this highlights our system's superior ability to generate intelligible long-form audio (significantly lower WER) while maintaining competitive speaker similarity.  We leave a more granular exploration of optimal architectures for this new task as a promising direction for future work
>
> | Model  | Sim-O (↑) | WER (↓)|
> |-|:-:|:-:|
> | Sesame | 0.53 | 2.71  |
> | Dia    | 0.54 | 3.10  |
> | Ours   | 0.53 | 1.81  |
>
>
> **Weakness-1_1. What kind of performance degradation occurs if the model is trained directly without the proposed curriculum learning? Demonstrating such differences could help highlight the effectiveness of the components proposed in the paper.**
>
> A2. Thank you for your question. We find that the proposed curriculum learning is essential for stable model convergence. Our preliminary experiments show that if we train the model directly on spontaneous data while skipping the audiobook data stage, the model fails to converge within 2000 steps. This result confirms the necessity of our curriculum learning approach. This initial stage builds an essential foundation, preparing the model for stable convergence on the more challenging spontaneous speech data.
>
> **Weakness-1_2.  Similarly, I am curious about the advantages of the interleaving approach you adopted compared to the turn-based interleaving method mentioned in line 163, particularly in terms of performance, sample quality, or perceptual (auditory) improvements.**
>
> A3. That is an interesting question. We did, in fact, compare these two interleaving approaches on a very small scale during our preliminary experiments. The results, detailed in the table below, show that while objective metrics like SIM-O and WER are comparable, our proposed full-text-to-full-audio interleaving approach yields subjectively superior results, particularly in terms of coherence and spontaneity.
>
> | Method | Spontaneity (↑) | Coherence (↑) | SIM-O (↑) | WER (%) (↓) |
> |-|:-:|:-:|:-:|:-:|
> | Turn-based Interleaving | 4.33 | 4.16 | **0.81** | **1.80** |
> | Our Full-text-to-full-audio Interleaving | **4.58** | **4.50** | 0.80 | 1.83 |
>
> We hypothesize two primary reasons for this advantage. First, given the strong text-to-speech mapping, the holistic nature of our interleaving approach inherently guides the model to capture **long-range, global dependencies**, as the corresponding text and speech segments are often distant from each other. In contrast, a turn-based method biases the model towards **local, turn-level dependencies**, which can result in less coherent output over an entire dialogue. Second, when generating the speech for any given turn, our interleaving approach allows the causal model to attend to the **complete** text transcript of the entire podcast. A turn-based method, however, provides only a **partial** context, as the model cannot attend to the text of subsequent turns. We believe this complete contextual view is a key factor in the improved coherence.

---

> > ### Comment · Reviewer_DLz7 · 2025-08-02
> >
> > Thank you for your kind response. My concerns and questions have been resolved by the authors, and I have adjusted the score accordingly.

---

> ### Author Response · Authors · 2025-08-04
>
> Thank you for the positive update. We're glad we could resolve your concerns. We truly appreciate your time and reconsideration

---

### Official Review · Reviewer_mbG2 · 2025-06-16

**Clarity:** 3
**Significance:** 3
**Originality:** 2
**Rating:** 4
**Confidence:** 4

**Summary:**

The paper introduces a long-form, high-quality, multispeaker podcast generation system, which can generate contextually coherent multi-turn long podcasts. They employ a DAC [Kumar et al., 2023] type discrete tokenizer over the 17th layer of pretrained W2vBert 2.0 [Schneider et al., 2019]. Then they use this encoded representation for further training a 2.5B llama model. They employed a data preprocessing pipeline [Yu et al., 2024] to prepare 500k hours of data. They employ a three-stage learning process where the first two stages are trained with a Chinese-only dataset, and the third stage is employed using English and Chinese datasets. The first stage is similar to training of Vall-E [Wang et al., 2023], the second stage is pretraining on chinese audiobooks and the last stage is pretraining on both english and chinese podcast data. The turns during the podcasts are interleaved with speaker ID tags to learn better turn-taking and multi-turn prosody. Finally, a causal speech de-tokenizer similar to CosyVoice 2 [Du et al., 2024b] was employed to transform speech representation to a mel-spectrogram, allowing for generating long audio chunks.

**Questions:**

- In Section 4.1.1 subheading - Sequence Design:
It is a bit unclear with the text where the speaker change token is added, it says
>Line 165: incorporate a special speaker change token after prompts and each podcast turn.

While in the concatenation $\mathcal{S}$  does not specifies how it is added Line 172-173.  Is it like in the Figure 2? (If yes please update this section)

- The speech tokeniser uses RVQGAN and generates 8 dimensional codes. How do you model these 8 dimensional codes with the LLM? Is it just the first dimension is modelled or all 8?

**Ethical Concerns:**

["NO or VERY MINOR ethics concerns only"]

**Final Justification:**

There are still questions around novelty and a little over-exaggeration of hyperparameter tuning. However, the added experimentation during the rebuttal period made me change the score, as initially the experiments were weaker and more details around them were required. I still feel that for the initial paper submission, the confidence intervals are hard to get, but the experiments done during the rebuttal seem to be adequate enough.

**Limitations:**

Yes

**Paper Formatting Concerns:**

Line 331 typo: speraker ->  speaker

Line 571 typo: fprocess -> process

**Quality:**

3

**Strengths And Weaknesses:**

Strengths:
* First open the weight models for multiturn multispeaker podcast generation
* High-quality listening examples improvements over concatenative or single-speaker baselines
* Uses ASR-based weaker labels as a proxy for spontaneous annotated data

Weakenesses:
- The paper lacks novelty; the data was preprocessed using [Yu et al., 2024] pipeline, the tokenizer was similar to [Kumar et al., 2023]. The spontaneous dataset was generated similarly to SLIDE [Lu et al., 2025]. Multistage training is a common approach to pretraining and fine-tuning on the required dataset, and the speech de-tokenizer is similar to CosyVoice 2 [Du et al., 2024b]. However, the authors do stitch them together very well and scale them to 500k hours of data.
- The authors make a novel assumption: "ASR transcripts can act as a partial proxy for speech spontaneity.". However, it is not formally validated; it would have been nice to quantify this assumption on a small subset for which they have ground truth text transcriptions to see how much of the spontaneity is retained after ASR transcription. This would validate this assumption.
- Subjective evaluations were performed by 10 raters, that is number is low for such long generations. Further, the author's evaluation of long podcasts is done similarly to short sentence evaluation, while this should not be the case, i.e, one number for a few minutes of audio. It should be performed over time; one way to do this was proposed in [1].
- Some evaluation details are lacking, like the question asked for subjective evaluation (especially for coherence), how the confidence intervals were calculated for subjective evaluation, as in Table 3, the confidence intervals are around 0.09, which seems to be hard to get with 10 raters.

[1] Park, S.J., Salazar, J., Jansen, A., Kinoshita, K., Ro, Y.M. and Skerry-Ryan, R.J., 2024. Long-Form Speech Generation with Spoken Language Models. _arXiv preprint arXiv:2412.18603_.

---

> ### Author Rebuttal · Authors · 2025-07-31
>
> First of all, we want to thank the reviewer for your careful reading and providing a lot of constructive comments! Below we address the concerns mentioned in the review.
>
> **Weaknesses-1. The paper lacks novelty; the data was preprocessed using [Yu et al., 2024] pipeline, the tokenizer was similar to [Kumar et al., 2023]. The spontaneous dataset was generated similarly to SLIDE [Lu et al., 2025]. Multistage training is a common approach to pretraining and fine-tuning on the required dataset, and the speech de-tokenizer is similar to CosyVoice 2 [Du et al., 2024b]. However, the authors do stitch them together very well and scale them to 500k hours of data.**
>
> A1.  We appreciate the reviewer's detailed feedback. While we build upon a strong foundation of prior research, we would like to clarify that our work is not merely an assembly of existing methods. We introduce several critical innovations at each stage to address the unique challenges of generating long-form, multi-speaker podcast speech.
> - Data Processing Pipeline: The pipeline from [Yu et al., 2024] is designed for single-speaker, short-form TTS, where low-quality segments can be filtered. This is not feasible for long-form dialogue, where discarding any segments would destroy conversational continuity. We therefore develop a novel, multi-stage diarization post-processing pipeline to enhance robustness. This includes 1) speaker cluster merging to handle the speaker fragmentation (i.e., assign multiple speaker labels to the same actual speaker). 2)  chunk-based reassignment to handle segments containing multiple speakers, and 3) segment merging to handle  impractical sequence lengths (shorter than 1s or longer than 100s).
> - Semantic Tokenizer: There is a fundamental distinction here. The tokenizer in [Kumar et al., 2023] operates on the raw audio waveform to produce **acoustic** tokens. In contrast, our tokenizer is designed to process self-supervised features to extract **semantic** tokens.
> - Spontaneous Speech Data: In fact, we did not **generate** any spontaneous dataset. Instead, we leverage the **raw** spontaneous speech and its corresponding ASR transcript from our large-scale raw data.
> - Multi-Stage Training: Our training strategy is not a general pretraining-and-finetuning schedule. It is a purpose-built curriculum learning where each stage addresses a distinct learning objective essential for conversational TTS: from foundational zero-shot ability, to long-term context modeling, and finally to the nuances of spontaneous speech.
> - Speech De-tokenizer: Our de-tokenizer is also distinct from that of CosyVoice 2, as it features a novel chunk-wise causal mask method that we develop specifically to accelerate the training of our architecture.
>
> These targeted innovations were integral to successfully scaling our model and achieving robust performance in the complex domain of multi-speaker conversational speech. While previous TTS research has mainly focused on short, single-speaker tasks, our model is one of the first academic works to tackle long-form, multi-speaker generation, highlighting the novelty of our contribution to this emerging field.
>
>
> **Weaknesses-2. The authors make a novel assumption: "ASR transcripts can act as a partial proxy for speech spontaneity.". However, it is not formally validated; it would have been nice to quantify this assumption on a small subset for which they have ground truth text transcriptions to see how much of the spontaneity is retained after ASR transcription. This would validate this assumption.**
>
> A2. Thanks for your excellent insights! To validate our novel assumption, we conduct a quantitative analysis. We select a one-hour subset of podcast audio and perform a manual transcription to obtain the GT transcript. Subsequently, we manually count the occurrences of spontaneous speech elements, including filler words, response words and repetitions & colloquialisms, in both the GT and the corresponding ASR transcripts. The results, detailed in the table below, show that the ASR output successfully retains a significant majority of these elements, providing strong evidence to validate our assumption.
>
>
> ||Filler Words|Response Words|Repetitions & Colloquialisms|
> |:---:|:---:|:---:|:---:|
> |GT Transcript|425|191|94|
> |ASR Transcript |359|164|88|
>
>
>
> **Weaknesses-3. Subjective evaluations were performed by 10 raters, that is number is low for such long generations. Further, the author's evaluation of long podcasts is done similarly to short sentence evaluation, while this should not be the case, i.e, one number for a few minutes of audio. It should be performed over time; one way to do this was proposed in [1].**
>
> A3. Thank you for your valuable feedback. We agree that evaluating long-form generation is a nuanced task. While the over-time method proposed in [1] is insightful, it is primarily designed for single-speaker scenarios. Our initial choice of assigning a single score to the entire podcast was primarily to mitigate the impact of speaker diarization errors during evaluation.
>
> However, we recognize the importance of the point you raised. To address this, we conduct a new evaluation with 10 knowledge sources and 20 raters on English podcast generation. In this round, the audio is segmented by speaking turns, and each turn is rated individually. As the results presented below indicate, this more granular, turn-by-turn evaluation aligns with the findings reported in our main paper.
>
> | | Spontaneity(↑) | Coherence(↑) | Intelligibility(↑) | Quality(↑) | Similarity(↑) | SIM-O(↑) | WER(↓) |
> |---|:---:|:---:|:---:|:---:|:---:|:---:|:---:|
> |Cosyvoice2| 3.75±0.04 | 3.80±0.04 | 4.55±0.03 | 4.35±0.04 | **4.54±0.02** | 0.76 | 2.53 |
> |Concat Baseline| 3.62±0.03 | 3.65±0.04 | 4.12±0.03 | 3.65±0.03 | 4.01±0.03 | **0.78** | 2.36 |
> |Our System| **4.60±0.02** | **4.53±0.03** | **4.61±0.02** | **4.38±0.02** | 4.22±0.04 | 0.56 | **1.69** |
>
>
> **Weaknesses-4. Some evaluation details are lacking, like the question asked for subjective evaluation (especially for coherence), how the confidence intervals were calculated for subjective evaluation, as in Table 3, the confidence intervals are around 0.09, which seems to be hard to get with 10 raters.**
>
> A4. Thanks for your question. We are happy to provide the following clarifications. For the subjective evaluation, raters were asked to assess the generated audio based on the following questions.
> - Spontaneity: Assesses how spontaneous the audio sounds. This focuses on the presence of common spoken elements (e.g., filler words such as 'um,' 'uh,' 'like,' 'you know,' 'so,' 'right?', and response words such as 'Yeah,' 'Right,' 'Okay') as well as a fluid and casual flow. Rated on a 1-5 scale with 0.5-point increments, where 5 is best.
> - Coherence: Evaluates the overall coherence of the audio. This includes the naturalness of transitions between speakers and whether the audio is globally consistent. Rated on a 1-5 scale with 0.5-point increments, where 5 is best.
> - Intelligibility: Measures whether the speech is clearly articulated and consistent with the provided text. Rated on a 1-5 scale with 0.5-point increments, where 5 is best.
> - Quality: Assesses the overall quality of the audio. Rated on a 1-5 scale with 0.5-point increments, where 5 is best.
> - Similarity: Determines if the speaker's voice is consistent with the voice in the provided reference audio. Rated on a 1-5 scale with 0.5-point increments, where 5 is best.
>
> Regarding the confidence intervals in Table 3, they are calculated by pooling together all individual ratings from all raters across every sample in the test set. We hope this clarifies our procedure.
>
>
> **Questions-1. In Section 4.1.1 subheading - Sequence Design: It is a bit unclear with the text where the speaker change token is added, it says
> Line 165: incorporate a special speaker change token after prompts and each podcast turn.
> While in the concatenation  does not specifies how it is added Line 172-173. Is it like in the Figure 2? (If yes please update this section)**
>
> A5. We apologize for the ambiguity in the original paper. We will revise the text in that section to explicitly reference the figure, ensuring the description of the concatenation process is clear. We appreciate you bringing this to our attention.
>
> **Questions-2. The speech tokeniser uses RVQGAN and generates 8 dimensional codes. How do you model these 8 dimensional codes with the LLM? Is it just the first dimension is modelled or all 8?**
>
> A6. Thank you for the question. There appears to be a misunderstanding based on the text in our paper, "The 1024-dimensional SSL feature is projected into an 8-dimensional space for quantization using an 8192-entry codebook." In fact,  we do not model 8-dimensional residual VQ codes or a 8-dimensional latent with the LLM. The "8-dimensional space" refers to the latent dimensionality of each vector within the codebook, not the output of the quantization process. Such expressions are consistent with previous works [1]. The quantizer itself outputs a single, discrete index from this codebook for each speech frame. Consequently, the LLM models the sequence of these discrete indices, not the 8-dimensional latent vectors.
>
> [1] Kumar, Rithesh, et al. "High-fidelity audio compression with improved rvqgan." Nips 2023.

---

> > ### Author Response · Authors · 2025-08-04
> >
> > With the discussion deadline approaching, we want to gently check in and see if our rebuttal was helpful in addressing your concerns. We are ready to provide any further clarification.

---

> > > ### Comment · Reviewer_mbG2 · 2025-08-05
> > > **Thank you for clarifying some points.**
> > >
> > > I would like to thank the authors for clarifying some points. Most of my comments have been answered. However, I have a few follow-up questions
> > >
> > > For A1:
> > >
> > > > multi-stage diarization post-processing pipeline to enhance robustness
> > > I understand, but I feel there is still a minor lack of the novelty element primarily because
> > > 1. The running of Pyannote for speaker diarization is a very common approach with found data [1] (Currently, there is no mention of clustering in your article. Please document this step in the article as well)
> > > 2. Part 2 is still similar to CosyVoice 2. They also employed a force aligner to filter data.
> > > 3. Part three seems to be heuristic around filtering small and large segments.
> > >
> > > But again, these parts together do play really well.
> > >
> > > > Semantic Tokenizer: There is a fundamental distinction here. The tokenizer in [Kumar et al., 2023] operates on the raw audio waveform to produce acoustic tokens. In contrast, our tokenizer is designed to process self-supervised features to extract semantic tokens.
> > >
> > > Thank you for clarifying. Could you please also shed light on how you disentangle the **semantic** component from the **acoustic** component? Or is it quantised MASKGCT?
> > >
> > >
> > > > Spontaneous Speech Data: In fact, we did not generate any spontaneous dataset. Instead, we leverage the raw spontaneous speech and its corresponding ASR transcript from our large-scale raw data.
> > >
> > > Thank you for the clarification.
> > >
> > > > Multi-Stage Training: Our training strategy is not a general pretraining-and-finetuning schedule. It is a purpose-built curriculum learning where each stage addresses a distinct learning objective essential for conversational TTS: from foundational zero-shot ability, to long-term context modeling, and finally to the nuances of spontaneous speech.
> > >
> > > Thank you again for the clarification. This makes it a bit clearer for me.
> > >
> > > > Speech De-tokenizer: Our de-tokenizer is also distinct from that of CosyVoice 2, as it features a novel chunk-wise causal mask method that we develop specifically to accelerate the training of our architecture.
> > >
> > > To my understanding, CosyVoice 2's detokenizer also employs a chunk-aware causal transformer, which is used in a similar manner for streaming purposes.
> > >
> > >
> > > A2:  Thank you, this resolves my concern for this question.
> > >
> > > A3: Thank you very much for the detailed evaluation. This makes the difference evident!
> > >
> > > For A4: I am still a little concerned about the confidence intervals. Could you give an overall summary of how many samples each rater evaluated, approximately?
> > >
> > >
> > > For the answer to Question 2: Thank you for the clarification. This makes it clearer for me.
> > >
> > > [1] Tanveer, M.I., Casabuena, D., Karlgren, J., Jones, R. (2022) Unsupervised Speaker Diarization that is Agnostic to Language, Overlap-Aware, and Tuning Free. Proc. Interspeech 2022, 1481-1485, doi: 10.21437/Interspeech.2022-10605

---

> ### Author Response · Authors · 2025-08-05
> **Further Clarification-Part1**
>
> We thank the reviewer mbG2 for carefully reading our responses during the rebuttal stage and for engaging in further discussion.
>
> Q1. multi-stage diarization post-processing pipeline to enhance robustness I understand, but I feel there is still a minor lack of the novelty element primarily because
>
> 1. The running of Pyannote for speaker diarization is a very common approach with found data [1] (Currently, there is no mention of clustering in your article. Please document this step in the article as well)
> 2. Part 2 is still similar to CosyVoice 2. They also employed a force aligner to filter data.
> 3. Part three seems to be heuristic around filtering small and large segments.
>
> A1. We appreciate that the reviewer acknowledges the effectiveness of our data pipeline. We apologize for omitting some details in the current manuscript and will revise the paper to include them. Regarding novelty, we would like to clarify the following:
>
> We acknowledge that *Pyannotate* has been used in the data pipelines of previous research works such as *Emilia* [1]. However, those works primarily focus on generating short and isolated audio segments (typically <30s). In contrast, building long-context conversational speech data with multiple speakers presents unique challenges: 1) Speaker Consistency: It is essential to ensure that speaker labels remain consistent across different utterances. This requirement is more stringent than in prior work, where it suffices that each short segment contains a single speaker.  2) Maintaining Contextual Integrity:  To preserve the semantic flow of conversations, we cannot simply filter out low-quality sentences as previous works often do.
>
> Because of these challenges, we cannot directly rely on the original *Pyannotate* annotations. Specifically:  *Pyannotate* segments audio using a Voice Activity Detector (VAD), followed by hierarchical clustering to assign speaker labels. However, speaker changes often do not coincide with silence, so VAD-based segmentation may fail to split utterances at speaker turns. This results in segments containing multiple speakers, which negatively affects generation stability.
>
> To mitigate this,  we lower the VAD threshold to make segmentation more fine-grained, reducing the likelihood of multiple speakers within one segment. On top of the *Pyannotate* clustering results, we merge clusters with cosine similarity > 0.65 to avoid assigning multiple IDs to the same speaker.
>
> We further perform intra-sentence clustering. When multiple cluster centers are detected within a single sentence, we do not discard the sentence (to maintain context), but split it at the point where the speaker embedding cluster label changes. We then recompute each segment’s speaker embedding using the updated cluster center to ensure label consistency.
>
> Additionally, we merge consecutive segments with high speaker similarity to provide each utterance with sufficient context.
>
> Through the above techniques, we obtain a high-quality conversational speech dataset. While we acknowledge that some of these methods may be considered engineering "tricks", they are crucial for effectively training long-context speech generation models. To the best of our knowledge, publicly available data pipelines and prior works on long-form multi-speaker modeling remain very limited.
>
> [1] He, Haorui, et al. "Emilia: An extensive, multilingual, and diverse speech dataset for large-scale speech generation."
>
> Q2. Could you please also shed light on how you disentangle the semantic component from the acoustic component? Or is it quantised MASKGCT?
>
> A2. Thanks for your questions. The paradigm of representing speech through separate semantic and acoustic tokens has gained significant traction in the TTS community [1, 2]. Specifically, acoustic tokens are discrete representations designed to capture fine-grained acoustic details for high-fidelity waveform reconstruction, typically learned through a direct reconstruction task. Conversely, semantic tokens provide a coarse, high-level representation that preserves essential linguistic content, such as phonetics and semantics, while abstracting away paralinguistic information like acoustic details and speaker identity. These semantic tokens are most commonly derived by quantizing features from self-supervised speech features.
>
> [1] Kharitonov, Eugene, et al. "Speak, read and prompt: High-fidelity text-to-speech with minimal supervision."
>
> [2] Borsos, Zalán, et al. "Audiolm: a language modeling approach to audio generation."

---

> ### Author Response · Authors · 2025-08-05
> **Further Clarification-Part2**
>
> Q3. To my understanding, CosyVoice 2's detokenizer also employs a chunk-aware causal transformer, which is used in a similar manner for streaming purposes.
>
> A3. That is a critical point. While both models utilize a chunk-aware transformer, our key innovation lies in the training methodology, enabled by a novel chunk-wise causal mask. This mask facilitates a more efficient training paradigm.
>
> To illustrate, consider a data sample of four chunks: $[x_0, x_1, x_2, x_3]$, where $x_i$ is a clean chunk and $\hat{x_i}$ is its corresponding masked/noisy version.
>
> As for CosyVoice 2, for each training sample, their method randomly selects a single split point. For instance, it might use $[x_0, x_1]$ as the clean prompt to predict the noisy target $[\hat{x_2}, \hat{x_3}]$. This means only one prompt-target configuration is learned from the sample in each forward pass.
>
> In contrast, our chunk-wise causal mask allows the model to learn from all possible prompt-target splits in parallel within a single data batch. For the same sample, our model concurrently optimizes the predictions for every scenario, including:
> - Predicting $[\hat{x_1}, \hat{x_2}, \hat{x_3}]$ given the prompt $[x_0]$
> - Predicting $[\hat{x_2}, \hat{x_3}]$ given the prompt $[x_0, x_1]$
> - Predicting $[\hat{x_3}]$ given the prompt $[x_0, x_1, x_2]$
>
> This parallel training approach makes fuller use of each data sample, leading to better training efficiency.
>
> Q4. Could you give an overall summary of how many samples each rater evaluated, approximately?
>
> A4. Thanks for your query. For the evaluation in the main paper, we recruit 10 raters to assess 3 systems. The test set comprised 4 long-form dialogue samples, and raters provided one evaluation per sample, meaning each rater evaluated a total of 12 samples (3 systems × 4 samples). This corresponds to a listening time of approximately 1.2 hours per rater. This resulted in 40 evaluations for each system.
>
> For the evaluation during the rebuttal period, we recruit 20 raters for the same 3 systems. The test set comprised 10 long-form dialogues with an average of 55 turns, and raters evaluated every turn. Thus, each rater evaluated approximately 1,650 turns (3 systems × 10 dialogues × 55 turns). This corresponds to a listening time of approximately 3.0 hours per rater. This resulted in 11,000 evaluations for each system.

---

> > ### Comment · Reviewer_mbG2 · 2025-08-05
> > **Thank you for interesting discussion**
> >
> > Thank you for an interesting discussion. Q2 was emphasised more on how **you** disentangled, not the general definition. But all the other concerns have been resolved by the authors, and I have adjusted my rating accordingly.

---

> > > ### Author Response · Authors · 2025-08-06
> > >
> > > Thank you for the positive update. We're glad we could resolve most of your concerns. We truly appreciate your time and reconsideration.
> > >
> > > Regarding Q2, we acknowledge that our work does not focus on decoupling. Instead, our intention here is to highlight the distinction between our tokenizer and the one used by Kumar et al. (2023). As detailed in Appendix A, our tokenizer is a VQ-VAE model. It is trained with the objective of reconstructing self-supervised speech features from W2v-BERT. Our experiments suggest that this VQ-VAE approach yields tokens with enhanced reconstructive quality for Chinese speech, particularly in prosody, when compared to simpler k-means quantization.

---

### Official Review · Reviewer_U3tD · 2025-07-03

**Clarity:** 3
**Significance:** 4
**Originality:** 3
**Rating:** 4
**Confidence:** 4

**Summary:**

The paper proposes a podcast-generation framework trained on a large-scale corpus of conversational speech, enabling spontaneous two-speaker dialogue synthesis. To model spontaneous dialogue, the system uses ASR transcripts, containing natural filler words such as “yeah,” “um,” and “uh", as a proxy for spontaneous scripts during training. At inference time, a large language model generates spontaneous dialogue scripts from knowledge sources such as URLs, PDFs, or plain text. These scripts serve as input for the audio model.

To support long-form podcast generation, the model leverages the long-context modeling capability of Transformer to produce text-to-semantic token sequences for audio segments up to 800 seconds. These semantic tokens are converted into mel-spectrograms using a chunk-wise causal flow-matching model, followed by a vocoder to generate waveforms. The text-to-semantic module is trained to generate both speakers' utterances using speaker turn tokens and speaker-specific transcripts. Because long-form spontaneous speech data is limited, the model is trained using a curriculum learning strategy. The full pipeline is trained on approximately 500,000 hours of data, and both subjective and objective evaluations show that it outperforms CosyVoice 2 and a simple concatenation baseline in generating spontaneous dialogues.

**Questions:**

To what extent is the large-scale training data necessary for generating spontaneous speech from ASR transcripts? Could similar results be achieved with less data if more advanced modeling were used?

Speaker turn tokens are inserted based entirely on diarization. Is there any empirical evidence showing that these tokens remain robust and effective in the autoregressive text-to-semantic module, especially in cases of noisy diarization or frequent speaker changes?

**Ethical Concerns:**

["NO or VERY MINOR ethics concerns only"]

**Final Justification:**

The authors have addressed my concerns regarding data scale and baseline comparisons. Including this analysis in the appendix would aid understanding. I maintain my score as "borderline-accept".

**Limitations:**

Yes

**Paper Formatting Concerns:**

.

**Quality:**

3

**Strengths And Weaknesses:**

**Strengths**

The model effectively reflects spontaneous characteristics in generated dialogues, producing speech that includes filler words and natural turn-taking.

The architecture is carefully designed to enable generation of long-form dialogues exceeding 10 minutes, and demo samples sound natural and coherent.

The use of LLM-based spontaneous script generation makes it possible to turn arbitrary knowledge sources into conversational audio, opening applications similar to systems like NotebookLM.

**Weaknesses**

The system requires a substantial amount of data and compute resources to train.

There may be a mismatch between the ASR transcripts used as training targets and the LLM-generated spontaneous scripts used at inference time, which is not addressed in the analysis.

Baseline comparisons are limited to CosyVoice 2 and simple concatenation; comparisons with existing dialogue-generation models such as Sesame or Dia are not included, which could further highlight the model’s advantages.

---

> ### Author Rebuttal · Authors · 2025-07-31
>
> First of all, we want to thank the reviewer for your careful reading and providing a lot of constructive comments! Below we address the concerns mentioned in the review.
>
> **Weaknesses-1. The system requires a substantial amount of data and compute resources to train.**
>
> A1.  Thanks for your question. We acknowledge that our model's substantial resource requirements present a challenge. This is a design trade-off motivated by the proven success of scaling in the zero-shot TTS domain, as shown by models like VALL-E and NaturalSpeech 2. To mitigate this barrier for the community, we plan to open-source our pre-trained model. This will provide a resource-efficient pathway for future works to build upon our work by fine-tuning on smaller datasets instead of training from scratch. By taking this step, we hope to make the benefits of our large-scale approach more accessible. While our current model aligns with the scaling paradigm, enhancing TTS performance under intrinsic resource constraints remains a crucial and open research question that we intend to explore in future work.
>
> **Weaknesses-2. There may be a mismatch between the ASR transcripts used as training targets and the LLM-generated spontaneous scripts used at inference time, which is not addressed in the analysis.**
>
> A2. It is an excellent point. We analyze this potential mismatch in Section 5.2.2 (Table 3), where we compare outputs from ASR transcripts (GT Script) with those from LLM-generated text (Spontaneous Script). Specifically,  the overall performance is highly consistent between the two settings. While the ASR transcripts yield better spontaneity,  the LLM-generated scripts offer better intelligibility. This demonstrates that the mismatch does not significantly degrade performance.
>
> To further bridge this training-inference gap, we incorporate ASR transcripts as demonstrations within the LLM prompt (see Appendix C). This strategy guides the language model to generate outputs that more closely align with the stylistic properties of our training data.
>
> **Weaknesses-3. Baseline comparisons are limited to CosyVoice 2 and simple concatenation; comparisons with existing dialogue-generation models such as Sesame or Dia are not included, which could further highlight the model’s advantages.**
>
> A3. We thank the reviewer for this valuable suggestion. We did not initially include Sesame (released March 13) or Dia (April 22), as they are considered concurrent work under the NeurIPS policy (post March 1). Furthermore, as these systems were released as engineering projects without formal academic publications, a rigorous methodological comparison was challenging at the time.
>
> However, we agree that this comparison is beneficial and will conduct a new evaluation following the reviewer's suggestion. A critical challenge in this comparison is that both Sesame and Dia have significant constraints on their context length (2048 and 3072 tokens, respectively), which is far shorter than what is required for generating long-form podcasts. To accommodate these baselines, we have to discard parts of the long-range context during inference.  The results shown in the table below demonstrate that while maintaining competitive Sim-O, our model achieves a substantially lower WER, highlighting its superior ability to generate intelligible long-form audio.
>
> | Model  | Sim-O (↑) | WER (↓)|
> |-|:-:|:-:|
> | Sesame | 0.53 | 2.71  |
> | Dia    | 0.54 | 3.10  |
> | Ours   | 0.53 | 1.81  |
>
>
> **Questions-1. To what extent is the large-scale training data necessary for generating spontaneous speech from ASR transcripts? Could similar results be achieved with less data if more advanced modeling were used?**
>
>
> A4. Thank you for this critical question. Our experiments provide a clear rationale for our focus on data scale. As shown in Table below, We observe that reducing the training set by a factor of ten results in a marked decline in speech spontaneity and coherence, a decline in speaker similarity and speech quality, and a minor decline in intelligibility.
>
> |  | Spontaneity (↑) | Coherence (↑) | Intelligibility (↑) | Quality (↑) | Similarity (↑) | SIM-O (↑) | WER (↓) |
> |---|:---:|:---:|:---:|:---:|:---:|:---:|:---:|
> | 1/10 Data | 4.18±0.14       | 4.23±0.15     | 4.58±0.13           | 4.18±0.12   | 4.12±0.15      | 0.50      | 1.94    |
> | Full Data | **4.54±0.16**       | **4.50±0.15**     | **4.61±0.12** |**4.30±0.10**   | **4.25±0.18**      | **0.53**      | **1.81**    |
>
>
>
> While advanced modeling has yet to offer a full substitute for large-scale data, we believe our plan to open-source the pre-trained model provides a practical way to mitigate this challenge. This will provide the community with a resource-efficient pathway, to fine-tune on smaller, high-quality datasets without the substantial cost of training from scratch.
>
> **Questions-2. Speaker turn tokens are inserted based entirely on diarization. Is there any empirical evidence showing that these tokens remain robust and effective in the autoregressive text-to-semantic module, especially in cases of noisy diarization or frequent speaker changes?**
>
> A5. Thanks for your insightful question. Our reported Word Error Rate (WER) provides direct empirical evidence for this. The WER is calculated on speaker turns that are segmented using these predicted speaker turn tokens. The low WER we achieve therefore serves as strong validation for their robustness, as significant errors in turn prediction would invariably lead to incorrect segmentation and a much higher error rate. Additionally, our data pipeline includes denoising and post-processing stages to mitigate noisy diarization issues.

---

> > ### Author Response · Authors · 2025-08-04
> >
> > With the discussion deadline approaching, we want to gently check in and see if our rebuttal was helpful in addressing your concerns. We are ready to provide any further clarification.

---

> > > ### Comment · Reviewer_U3tD · 2025-08-05
> > >
> > > Thanks to the authors for the clarification. The results regarding model performance across different data scales and comparisons with stronger dialogue-generation baselines have been well addressed. Including this content in the appendix could help improve readers’ understanding. Releasing the model would also be valuable in supporting the community. I will maintain my original score.

---

> ### Author Response · Authors · 2025-08-05
>
> We are very grateful for your thoughtful feedback and time spent on reviewing. We are pleased that our additional experiments addressed the main concerns. Following your valuable suggestion, we will incorporate these results into the appendix. Thank you again for your support of our work.

---

### Author Response · Authors · 2025-08-07

We are grateful to the Area Chair and all reviewers for a highly constructive discussion period. We are pleased to see a consensus emerging, with all reviewers acknowledging that their major concerns have been resolved and expressing a positive attitude towards our work.

---

### Decision · Program_Chairs · 2025-09-17

**Decision:**

Accept (poster)

**Comment:**

**Summary**

The paper proposes a podcast-generation framework trained on a large-scale corpus of conversational speech, enabling spontaneous two-speaker dialogue synthesis.

**Reasons to accept**

This presents the first open-weight  multi-turn multi-speaker TTS model.  The experimental results and examples were convincing to the reviewers.

**Reasons not to accept**

There is some concern over the novelty of the work (mbG2).  While the components have note put together in this way before, many of the technical pieces have clear origins in other works.

**Decision rationale**

While the reviewers did not express outright enthusiasm to accept this paper, the written reviews were consistently positive.  The weaknesses (unaddressed by the discussion period) were relatively minor.  GIven the novelty of application and open-source contribution, I would advocate for this paper being accepted.  There is sufficient innovation (novelty and significance) without substantial enough weaknesses to reject.